# A Call to Reflect on Evaluation Practices for Failure Detection in Image Classification

**Paul F. Jaeger[1,2], Carsten T. Lüth[1,2], Lukas Klein[1,2,3] & Till J. Bungert[1,2]**

[1]Interactive Machine Learning Group, German Cancer Research Center, Heidelberg, Germany
[2]Helmholtz Imaging, DKFZ, Heidelberg, Germany
[3]Institute for Machine Learning, ETH Zürich, Zürich, Switzerland

`p.jaeger@dkfz-heidelberg.de`

## Abstract

Reliable application of machine learning-based decision systems in the wild is one of the major challenges currently investigated by the field. A large portion of established approaches aims to detect erroneous predictions by means of assigning confidence scores. This confidence may be obtained by either quantifying the model's predictive uncertainty, learning explicit scoring functions, or assessing whether the input is in line with the training distribution. Curiously, while these approaches all state to address the same eventual goal of detecting failures of a classifier upon real-world application, they currently constitute largely separated research fields with individual evaluation protocols, which either exclude a substantial part of relevant methods or ignore large parts of relevant failure sources. In this work, we systematically reveal current pitfalls caused by these inconsistencies and derive requirements for a holistic and realistic evaluation of failure detection. To demonstrate the relevance of this unified perspective, we present a large-scale empirical study for the first time enabling benchmarking confidence scoring functions w.r.t. all relevant methods and failure sources. The revelation of a simple softmax response baseline as the overall best performing method underlines the drastic shortcomings of current evaluation in the abundance of publicized research on confidence scoring. Code and trained models are at `https://github.com/IML-DKFZ/fd-shifts`.

## 1 Introduction

*"Neural network-based classifiers may silently fail when the test data distribution differs from the training data. For critical tasks such as medical diagnosis or autonomous driving, it is thus essential to detect incorrect predictions based on an indication of whether the classifier is likely to fail"*.

Such or similar mission statements prelude numerous publications in the fields of misclassification detection (MisD) (Corbière et al., 2019; Hendrycks and Gimpel, 2017; Malinin and Gales, 2018), Out-of-Distribution detection (OoD-D) (Fort et al., 2021; Winkens et al., 2020; Lee et al., 2018; Hendrycks and Gimpel, 2017; DeVries and Taylor, 2018; Liang et al., 2018), selective classification (SC) (Liu et al., 2019; Geifman and El-Yaniv, 2019; 2017), and predictive uncertainty quantification (PUQ) (Ovadia et al., 2019; Kendall and Gal, 2017), hinting at the fact that all these approaches aim towards the same eventual goal: Enabling safe deployment of classification systems by means of *failure detection*, i.e. the detection or filtering of erroneous predictions based on *ranking of associated confidence scores*. In this context, any function whose continuous output aims to separate a classifier's failures from correct predictions can be interpreted as a *confidence scoring function* (CSF) and represents a valid approach to the stated goal. This holistic perspective on failure detection reveals extensive shortcomings in current evaluation protocols, which constitute major bottlenecks in progress toward the goal of making classifiers suitable for application in real-world scenarios. Our work is an appeal to corresponding communities to reflect on current practices and provides a technical deduction of a unified evaluation protocol, a list of empirical insights based on a large-scale study, as well as hands-on recommendations for researchers to catalyze progress in the field.

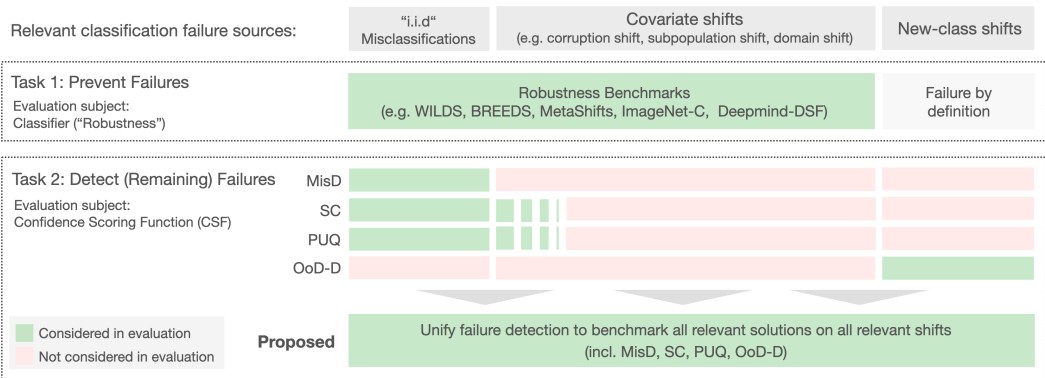

Figure 1: **Holistic perspective on failure detection.** Detecting failures should be seen in the context of the overarching goal of *preventing silent failures of a classifier*, which includes two tasks: preventing failures in the first place as measured by the "robustness" of a classifier (Task 1), and detecting the non-prevented failures by means of CSFs (Task 2, focus of this work). For *failure prevention* across distribution shifts, a consistent task formulation exists (featuring accuracy as the primary evaluation metric) and various benchmarks have been released covering a large variety of realistic shifts (e.g. image corruption shifts, sub-class shifts, or domain shifts). In contrast, progress in the subsequent task of *detecting* the non-prevented failures by means of CSFs is currently obstructed by three pitfalls: 1) A diverse and inconsistent set of evaluation protocols for CSFs exists (MisD, SC, PUQ, OoD-D) impeding comprehensive competition. 2) Only a fraction of the spectrum of realistic distribution shifts and thus potential failure sources is covered diminishing the practical relevance of evaluation. 3) The task formulation in OoD-D fundamentally deviates from the stated purpose of detecting classification failures. Overall, the holistic perspective on failure detection reveals an obvious need for a unified and comprehensive evaluation protocol, in analogy to current robustness benchmarks, to make classifiers fit for safety-critical applications. Abbreviations: CSF: Confidence Scoring Function, OoD-D: Out-of-Distribution Detection, MisD: Misclassification Detection, PUQ: Predictive Uncertainty Quantification, SC: Selective Classification.

## 2 PITFALLS OF CURRENT EVALUATION PRACTICES

Figure 1 gives an overview of the current state of *failure detection* research and its relationship to the preceding *failure prevention* task, which is measured by classifier robustness. This perspective reveals three main pitfalls, from which we derive three requirements *R1-R3* for a comprehensive and realistic evaluation in failure detection:

**Pitfall 1: Heterogeneous and inconsistent task definitions.** To achieve a meaningful evaluation, all relevant solutions toward the stated goal must be part of the competition. In research on failure detection, currently, four separate fields exist each evaluating proposed methods with their individual metrics and baselines. Incomplete competition is first and foremost an issue of historically evolved delimitations between research fields, which go so far that employed metrics are by design restricted to certain methods. **MisD:** Evaluation in MisD (see Section B.2.1 for a formal task definition) exclusively measures *discrimination* of a classifier's success versus failure cases by means of ranking metrics such as $\mathrm{AUROC_f}$[1] (Hendrycks and Gimpel, 2017; Jiang et al., 2018; Corbière et al., 2019; Bernhardt et al., 2022). This protocol excludes a substantial part of relevant CSFs from comparison, because any CSF that affects the underlying classifier (e.g. by introducing dropout or alternative loss functions) alters the set of classifier failures, i.e. ground truth labels, and thus creates their individual test set (for a visualization of this pitfall see Figure 4). As an example, a CSF that negatively affects the accuracy of a classifier might add easy-to-detect failures to its test set and benefit in the form of high $\mathrm{AUROC_f}$ scores. As depicted in Figure 1, we argue that the task of detecting failures is no self-purpose, but *preventing* and *detecting* failures are two sides of the same coin when striving to avoid silent classification failures. Thus, CSFs should be evaluated as part of a symbiotic system with the associated classifier. While additionally reporting classifier accuracy associated with each CSF

---

[1]where the "f" denotes "failure" (see Appendix B for details), not to be confused with the classification AUROC.

renders these effects transparent, it requires nontrivial weighting of the two metrics when aiming to rank CSFs based on a single score. **PUQ:** Research in PUQ often remains vague about the concrete application of extracted uncertainties stating the purpose of "meaningful confidence values" (Ovadia et al., 2019; Lakshminarayanan et al., 2017) (see Appendix B.2.3 for a formal task definition), which conflates the related but independent use-cases of *failure detection* and *confidence calibration*. This (arguably vague) goal is reflected in the evaluation, where typically strictly proper scoring rules (Gneiting and Raftery, 2007) such as negative log-likelihood assess a combination of *ranking* and *calibration* of scores. However, for failure detection use cases, arguably an explicit assessment of failure detection performance is desired (see Appendix C for a discussion on how calibration relates to failure detection). Furthermore, these metrics are specifically tailored towards probabilistic predictive outputs such as softmax classifiers and exclude all other CSFs from comparison.

→  *Requirement 1 (R1): Comprehensive evaluation requires a single standardized score that applies to arbitrary CSFs while taking into account their effects on the classifier.*

**Pitfall 2: Ignoring the major part of relevant failure sources.** As stated in the introductory quote, research on failure detection typically expects classification failures to occur when inputs upon application differ from the training data distribution. As shown in Figure 1, we distinguish "covariate shifts" (label-preserving shifts) versus "new-class shifts" (label-altering shifts). For a detailed formulation of different failure sources, see Appendix A. The fact that in the related task of *preventing failures* a myriad of nuanced covariate shifts have been released on various data sets and domains (Koh et al., 2021; Santurkar et al., 2021; Hendrycks and Dietterich, 2019; Liang and Zou, 2022; Wiles et al., 2022) to catalyze real-world progress of classifier robustness begs the question: If simulating realistic classification failures is such a delicate and extensive effort, why are there no analogous benchmarking efforts in the research on *detecting failures*? In contrast, CSFs are currently almost exclusively evaluated on i.i.d. test sets (**MisD, PUQ, SC**). Exceptions (see hatched areas in Figure 1) are a PUQ study that features corruption shifts (Ovadia et al., 2019), or SC evaluated on sub-class shift (comparing a fixed CSF under varying classifiers) (Tran et al., 2022) and applied to question answering under domain shift (Kamath et al., 2020). Further, research in **OoD-D** (see Section B.2.2 for a formal task definition) exclusively evaluates methods under one limited fraction of failure sources: new-class shifts (see images 7 and 8 in Figure 2 (right panel)). A recent trend in this area is to focus on "near OoD" scenarios, i.e. shifts affecting semantic image features but leaving the context unchanged (Winkens et al., 2020; Fort et al., 2021; Ren et al., 2021). While the notion that nuanced shifts might bear more practical relevance compared to vast context switches seems reasonable, the term "near" is misleading, as it ignores the whole spectrum of even "nearer" and thus potentially more relevant covariate shifts, which OoD-D methods are not tested against. We argue that for most applications, it is not realistic to exclusively assume classification failures from label-altering shifts and no failures caused by label-preserving shifts.

→  *Requirement 2 (R2): Analogously to robustness benchmarks, progress in failure detection requires to evaluate on a nuanced and diverse set of failure sources.*

**Pitfall 3: Discrepancy between the stated purpose and evaluation.** The described limitations of **OoD-D** evaluation are only symptoms of a deeper rooted problem: Methods are not tested to predict failures of a classifier, but instead to predict an external, i.e. classifier-agnostic "outlier" label. In some cases, this formulation reflects the inherent nature of the given problem, such as in anomaly detection, where no underlying task is defined and the data sets are potentially unlabeled (Ruff et al., 2021). However, the majority of work on OoD-detection comes with a defined classification task, including training labels and states detecting failures of the classifier as their main purpose (Fort et al., 2021; Winkens et al., 2020; Lee et al., 2018; Hendrycks and Gimpel, 2017; DeVries and Taylor, 2018; Liang et al., 2018). Yet, this line of work falls short of justifying why associated methods are subsequently not shown to detect the said failures but are instead tested on the surrogate task of detecting distribution shifts in the data. Figure 2 shows that the outlier label constitutes a poor tool to define which cases we wish to filter because the question "what is an outlier?" is highly subjective for covariate shifts (see purple question marks). The ambiguity of the label extends to the concept of "inliers" (what extent of data variation is still considered i.i.d.?), which the protocol rewards to retain irrespective of whether they cause the classifier to fail (see purple lightning).

→  *Requirement 3 (R3): If there is a defined classifier whose incorrect predictions are to be detected, its respective failure information should be used to assess CSFs w.r.t the stated purpose instead of a surrogate task such as distribution shift detection.*

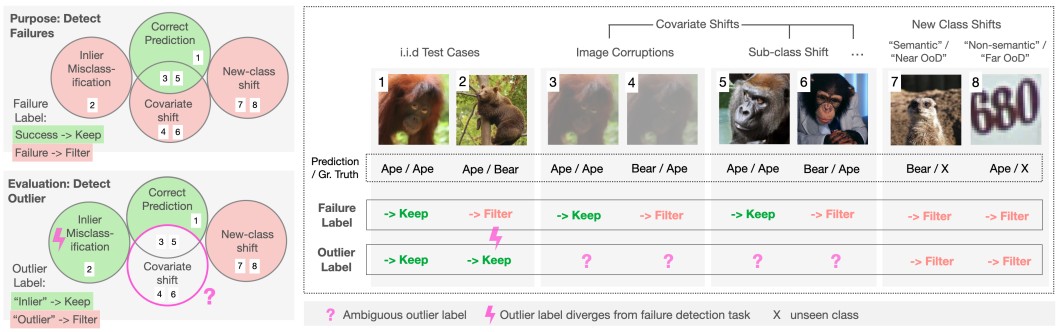

Figure 2: **Left: The discrepancy between the commonly stated purpose and evaluation in OoD-Detection.** The stated eventual purpose of detecting incorrect predictions of a classifier is represented by the binary "failure label" and its associated event space (top). However, in practice this goal is merely approximated by instead evaluating the detection of distribution shift, i.e. separating cases according to a binary "outlier label" irrespective of the classifier's correctness (bottom). **Right: Exemplary failure detection study under different types of failure sources.** A hypothetical classifier trained to distinguish "ape" from "bear" is evaluated on 8 images under a whole range of relevant distribution shifts: For instance, images 5 and 6 depict apes, but these were not among the breeds in the training data and thus constitute sub-class shifts. Images 7 and 8 depict entirely unseen categories, but while "meerkat" stays within the task context ("semantic", "near OoD"), "house number" represents a vast context switch ("non-semantic", "far OoD"). See Appendix A for a detailed formulation of shifts.

## 3 UNIFIED TASK FORMULATION

Parsing the quoted purpose statement at the start of Section 1 results in the following task formulation: Given a data set $\{(x_i, y_{\text{cl},i})\}_{i=1}^N$ of size $N$ with $(x_i, y_{\text{cl},i})$ independent samples from $\mathcal{X} \times \mathcal{Y}$ and $y_{\text{cl}}$ the class label, and given a pair of functions $(m, g)$, where $g : \mathcal{X} \to \mathbb{R}$ is a CSF and $m(\cdot, w) : \mathcal{X} \to \mathcal{Y}$ is the classifier including model parameters $w$, the classification output after failure detection is defined as:

$$(m, g)(x) := \begin{cases} P_m(y_{\text{cl}}|x, w), & \text{if } g(x) \geq \tau \\ \text{filter}, & \text{otherwise.} \end{cases} \tag{1}$$

Filtering ("detection") is triggered when $g(x)$ falls below a threshold $\tau$. In order to perform meaningful failure detection, a CSF $g(x)$ is required to output high confidence scores for correct predictions and low confidence scores for incorrect predictions based on the binary failure label

$$y_{\text{f}}(x, w, y) = \mathbb{I}(y_{\text{cl}} \neq \hat{y}_m(x, w)), \tag{2}$$

where $\hat{y}_m = \text{argmax}_{c \in \mathcal{Y}} P_m(y_{\text{cl}} = c|x, w)$ and $\mathbb{I}$ is the identity function (1 for true events and 0 for false events).

Despite accurately formalizing the stated purpose of numerous methods from MisD, OoD-D, SC, and PUQ, and allowing for evaluation of arbitrary CSFs $g(x)$, this generic task formulation is currently only stated in SC research (see Appendix B for a detailed technical description of all protocols considered in this work). To deduct an appropriate evaluation metric for the formulated task, we start with the ranking requirement on $g(x)$, which is assessed e.g. by $\text{AUROC}_{\text{f}}$ in MisD leading to Pitfalls described in Section 2. Following *R1* and modifying $\text{AUROC}_{\text{f}}$ to take classifier performance into account lets us naturally converge (see Appendix B.2.5 for the technical process) to a metric that has previously been proposed in SC as a byproduct, but is not widely employed for evaluation (Geifman et al., 2019): the Area under the-Risk-Coverage-Curve (AURC, see Equation 31). We propose to use AURC as the primary metric for all methods with the stated purpose of failure detection, as it fulfills all three requirements *R1-R3* in a single score. The inherently joint assessment of classifier accuracy and CSF ranking performance comes with a meaningful weighting between the two aspects, eliminating the need for manual (and potentially arbitrary) score aggregation. AURC measures the risk or error rate $(1 - \text{Accuracy})$ on the non-filtered cases averaged over all filtering thresholds (score range: [0,1], lower is better) and can be interpreted as directly assessing *the rate of silent failures occurring in a classifier*. While this metric enables a general evaluation of CSFs, depending on the use case, a more specific assessment of certain coverage regions (i.e. the ratio of remaining cases

after filtering) or even single risk-coverage working points might be appropriate. In Appendix F we provide an open source implementation of AURC fixing several shortcomings of previous versions.

## 3.1 Required modifications for current protocols

The general modifications necessary to shift from current protocols to a comprehensive and realistic evaluation of failure detection, i.e. to fulfill requirements *R1-R3*, are straightforward for considered fields (**SC**, **MisD**, **PUQ**, **OoD-D**): Researchers may simply consider reporting performance in terms of AURC and benchmark proposed methods against relevant baselines from all previously separated fields as well as on a realistic variety of failure sources (i.e. distribution shifts).

An additional aspect needs to be considered for **SC**, where the task is to solve *failure prevention and failure detection* at the same time (see Task 1 and Task 2 in Figure 1), i.e. the goal is to minimize *absolute* AURC scores. This setting includes studies that compare different classifiers while fixing the CSF (Tran et al., 2022). On the contrary, the evaluation of failure detection implies a focus on the performance of CSFs (Task 2 in Figure 1) while monitoring the classifier performance as a requirement (*R1*) to ensure a fair comparison of arbitrary CSFs. This shift of focus is reflected in the fact that the classifier architecture, as well as training procedure, are to be fixed (with some exceptions as described in Appendix E.4) across all compared CSFs. This way, external variations in classifier configuration are removed as a nuisance factor from CSF evaluation and the direct effect of the CSF on the classifier training is isolated to enable a *relative* comparison of AURC scores.

For evaluation of new-class shifts (as currently performed in **OoD-D**), a further modification is required: The current OoD-D protocol rewards CSFs for not detecting inlier misclassifications (see Figure 2). On the other hand, penalizing CSFs for not detecting these cases (as handled by AURC) would dilute the desired evaluation focus on new-class shifts. Thus, we propose to remove inlier misclassifications from evaluation when reporting a CSF's performance under new-class shift. Figure 5 visualizes the proposed modification. Notably, the proposed protocol does still consider the CSF's effect on classifier performance (i.e. does not break with *R1*), since higher classifier accuracy will remain to cause higher AURC scores (see Equations 29-31).

## 3.2 Own contributions in the presence of Selective Classification

Given the fact that the task definition in Equation 1 as well as AURC, the metric advocated for in this paper, have been formulated before in SC literature (see Appendix B.2.4 for technical details on current evaluation in SC)), it is important to highlight that the relevance of our work is *not* limited to advancing research in SC, but, next to the shift of focus on CSFs described in Section 3.1, we articulate a call to other communities (MisD, OoD-D, PUQ) to reflect on current practices. In other words, the relevance of our work derives from providing evidence for the necessity of the SC protocol in previously separated research fields and from extending their scope of evaluation (including the current scope of SC) w.r.t. compared methods and considered failure sources.

## 4 Empirical Study

To demonstrate the relevance of a holistic perspective on failure detection, we performed a large-scale empirical study, which we refer to as *FD-shifts* [2]. For the first time, state-of-the-art CSFs from MisD, OoD-D, PUQ, and SC are benchmarked against each other. And for the first time, analogously to recent robustness studies, CSFs are evaluated on a nuanced variety of distribution shifts to cover the entire spectrum of failure sources.

### 4.1 Utilized Data Sets

Appendix E features details of all used data sets, and Appendix A describes the considered distribution shifts. FD-Shift benchmarks CSFs on CAMELYON-17-Wilds (Koh et al., 2021), iWildCam-2020-Wilds (Koh et al., 2021), and BREEDS-ENTITY-13 (Santurkar et al., 2021), which have originally been proposed to evaluate classifier robustness (Task 1 in Figure 1) under sub-class shift in various domains. Further sub-class shifts are considered in the form of super-classes of CIFAR-100 (Krizhevsky,

---

[2]Code is at: `https://github.com/IML-DKFZ/fd-shifts`

2009), where one random class per super-class is held out during training. For studying corruption shifts, we report results on the 15 corruption types and 5 intensity levels proposed by Hendrycks and Dietterich (2019) based on CIFAR-10 as well as CIFAR-100. Regarding new-class shifts, we test on SVHN (Netzer et al., 2011), CIFAR-10/100 and TinyImagenet (Le and Yang, 2015) in a rotating fashion while considering the shift between CIFAR data sets as *semantic* and other shifts as *non-semantic*. Finally, we create additional semantic new-class shift scenarios by testing on held-out training classes (randomly sampled $40\%$ of all training classes) on SVHN and iWildCam-2020-Wilds.

## 4.2 Compared Methods

We compare the following CSFs: The maximum softmax response (MSR) calculated from the classifier's softmax output. **PUQ:** Predictive entropy based on softmax output (PE) and three predictive uncertainty measures based on Monte Carlo Dropout (MCD) (Gal and Ghahramani, 2016): mean softmax (MCD-MSR), predictive entropy (MCD-PE), and expected entropy (MCD-EE) (for technical formulations see Appendix I). For MCD we take 50 samples at test time. **MisD:** We include ConfidNet (Corbière et al., 2019), which is trained as an extension to the classifier and uses its regressed true class probability as a CSF. **SC:** We include DeepGamblers (DG), which uses loss attenuation based on portfolio theory to learn a confidence-like reservation score (DG-RES) (Liu et al., 2019). As the loss attenuation of DG's training paradigm might have a positive effect on the classifier itself, we additionally evaluate the softmax output (DG-MCD-MSR). **OoD-D:** We include the work of DeVries and Taylor (2018) [3]. Notably, ConfidNet, DG, and the work by Devries et al. are excellent examples of artificial separation of previous evaluations, as all three have not been compared before, despite their strong conceptual and technical similarities. We evaluate Maximum Logit Scores (MLS) as proposed by (Vaze et al., 2022) for semantic new-class shifts who argue that the softmax operation cancels out feature magnitudes relevant for OoD-D (we also add MLS scores averaged over MCD samples to the benchmark: MCD-MLS). Finally, we include the recently reported state-of-the-art approach: Mahalanobis Distance (MAHA) measured on representations of a Vision Transformer (ViT) that has been pretrained on ImageNet (Fort et al., 2021). **Classifiers:** Because this change in the classifier biases the comparison of CSFs, we additionally report the results for selected CSFs when trained in conjunction with a ViT classifier. For implementation and training details, see Appendix E. Since drawing conclusions from re-implemented baselines has to be taken with care, we report reproducibility results for all baselines including justifications for all hyperparameter deviations from the original configurations in Appendix J.

## 4.3 Results

The broad scope of this work reflects in the type of empirical observations we make: We view the holistic task protocol as an enabler for future research, thus we showcase the variety of research questions and topics that are now unlocked rather than providing an in-depth analysis on a single observation. Appendix G.1 features a discussion on how this study empirically confirms R1-R3 stated in Section 2.

Table 1 shows the results of the FD-Shifts benchmark measured as AURC scores. The reproducibility study in Appendix J confirms that none of the observed effects are caused by faulty re-implementations.

**None of the evaluated methods from literature beats the simple Maximum Softmax Response baseline across a realistic range of failure sources.** For both classifiers (CNN and ViT) the softmax baselines (either MSR or MCD-MSR) show the best or close to the best performance on all i.i.d. test sets and all shifts except new class shifts [4]. This is surprising given the claims of literature in MisD, SC, and OoD-D: All three tested methods based on the CNN-classifier (DG-Res, Devries, and ConfidNet) fail to generalize beyond the scenarios they have been proposed on, i.e. to more complex data sets (like iWildCam or BREEDS) and covariate distribution shifts (corruptions and sub-class

---

[3]We aimed to further include OpenHybrid (Zhang et al., 2020), but were not able to reproduce their results despite running the original code.

[4]Notably, the loss attenuation of DG seems to have positive effects on the softmax for i.i.d. settings leading to DG-MCD-MSR being the top-performing i.i.d. method with the CNN classifier on 3 out of 6 data sets.

Table 1: **FD-Shifts benchmark results measured as AURC $*10^3$ (score range:[0, 1000], lower is better ↓).** The color heatmap is normalized per column and classifier (separately for CNN and ViT), while whiter colors depict better scores. "cor" is the average over 5 intensity levels of image corruption shifts. AURC scores are averaged over 5 runs on all data sets with exceptions for the CNN: 10 runs on CAMELYON-17-Wilds (due to high volatility in results) and 2 runs on BREEDS. A second version of this table featuring CSF rankings is provided as Table 11. Abbreviations: ncs: new-class shift (s for semantic, ns for non-semantic), iid: independent and identically distributed, sub: sub-class shift, cor: image corruptions, c10/100: CIFAR-10/100, ti: TinyImagenet

| | | iWildCam | | | BREEDS | | CAMELYON | | CIFAR-100 | | | | | | | CIFAR-10 | | | | | SVHN | | | | |
|---|---|---|---|---|---|---|---|---|---|---|---|---|---|---|---|---|---|---|---|---|---|---|---|---|---|
| | study | iid | sub | s-ncs | iid | sub | iid | sub | iid | sub | cor | s-ncs | ns-ncs c10 | svhn | ti | iid | cor | s-ncs c100 | ns-ncs svhn | ti | iid | s-ncs | ns-ncs c10 | c100 | ti |
| MSR | CNN | 62.3 | 69.5 | 217 | 8.19 | 175 | 10.1 | 143 | 70.2 | 230 | 289 | 312 | 533 | | 321 | 5.62 | 93.0 | 226 | 429 | 198 | 4.85 | 132 | 55.1 | 55.8 | 55.7 |
| MLS | CNN | 90.2 | 87.8 | 240 | 11.7 | 188 | 10.1 | 143 | 87.6 | 233 | 312 | 318 | 544 | | 270 | 6.38 | 94.0 | 221 | 421 | 191 | 5.49 | 131 | 52.2 | 53.0 | 52.9 |
| PE | CNN | 62.7 | 69.9 | 215 | 8.21 | 174 | 10.1 | 143 | 77.0 | 234 | 308 | 319 | 587 | | 289 | 5.59 | 92.1 | 225 | 427 | 196 | 4.85 | 132 | 54.3 | 55.0 | 54.9 |
| MCD-MSR | CNN | 52.6 | 65.9 | 180 | 7.79 | 173 | 6.93 | 151 | 67.0 | 217 | 269 | 311 | 546 | | 318 | 5.23 | 81.4 | 226 | 438 | 203 | 4.75 | 164 | 54.1 | 54.9 | 54.0 |
| MCD-MLS | CNN | 84.1 | 78.6 | 189 | 11.5 | 186 | 7.22 | 156 | 79.0 | 220 | 279 | 315 | 528 | | 298 | 6.25 | 84.0 | 221 | 427 | 193 | 5.37 | 161 | 51.7 | 52.4 | 52.1 |
| MCD-PE | CNN | 53.5 | 67.1 | 175 | 7.89 | 173 | 6.93 | 151 | 72.4 | 217 | 273 | 312 | 540 | | 306 | 5.43 | 81.3 | 224 | 434 | 199 | 4.82 | 163 | 52.8 | 53.5 | 52.7 |
| MCD-EE | CNN | 54.1 | 67.6 | 175 | 7.94 | 173 | 6.94 | 151 | 72.5 | 218 | 274 | 313 | 544 | | 308 | 5.49 | 80.6 | 223 | 429 | 196 | 4.85 | 162 | 52.5 | 53.3 | 52.8 |
| MCD-MI | CNN | 53.5 | 71.0 | 218 | 8.19 | 176 | 7.40 | 164 | 76.9 | 221 | 282 | 313 | 535 | | 302 | 5.71 | 88.2 | 227 | 453 | 212 | 4.91 | 165 | 54.5 | 55.3 | 53.7 |
| ConfidNet | CNN | 143 | 144 | 214 | 8.30 | 176 | 5.04 | 132 | 72.7 | 232 | 290 | 321 | 552 | | 285 | 5.32 | 88.8 | 224 | 427 | 197 | 4.81 | 132 | 55.4 | 56.0 | 55.8 |
| DG-MCD-MSR | CNN | 54.8 | 72.7 | 224 | 7.05 | 167 | 4.30 | 273 | 66.3 | 216 | 268 | 311 | 547 | | 327 | 5.29 | 82.3 | 230 | 443 | 209 | 4.63 | 118 | 54.4 | 55.1 | 54.4 |
| DG-Res | CNN | 101 | 88.8 | 246 | 11.0 | 185 | 4.08 | 218 | 89.8 | 376 | 311 | 325 | 520 | | 291 | 5.39 | 88.8 | 243 | 422 | 195 | 5.68 | 124 | 52.4 | 53.1 | 53.4 |
| Devries et al. | CNN | 95.8 | 100 | 234 | 9.71 | 179 | 33.8 | 282 | 91.3 | 241 | 327 | 332 | 617 | | 338 | 5.22 | 97.5 | 226 | 420 | 195 | 7.27 | 154 | 55.7 | 57.7 | 55.4 |
| | | | | | | | | | | | | | | | | | | | | | | | | | |
| MSR | ViT | 86.1 | 70.6 | 267 | 1.89 | 105 | 0.09 | 62.5 | 14.2 | 70.0 | 69.0 | 238 | 456 | | 255 | 0.95 | 9.77 | 192 | 404 | 208 | 4.02 | 147 | 52.7 | 53.2 | 52.0 |
| MLS | ViT | 111 | 88.4 | 271 | 3.49 | 120 | 0.13 | 61.0 | 23.5 | 86.1 | 79.0 | 226 | 452 | | 253 | 1.27 | 10.4 | 192 | 401 | 205 | 4.91 | 146 | 51.2 | 52.0 | 50.2 |
| PE | ViT | 85.8 | 70.5 | 266 | 1.89 | 105 | 0.09 | 62.5 | 14.2 | 70.0 | 68.6 | 236 | 455 | | 254 | 0.95 | 9.75 | 192 | 404 | 208 | 4.02 | 147 | 52.5 | 53.1 | 51.8 |
| MCD-MSR | ViT | 113 | 145 | 163 | 1.54 | 108 | 0.05 | 304 | 13.7 | 76.3 | 91.1 | 240 | 490 | | 232 | 0.86 | 18.2 | 196 | 410 | 198 | 3.70 | 136 | 52.4 | 52.7 | 51.3 |
| MCD-MLS | ViT | 143 | 156 | 173 | 2.61 | 123 | 0.10 | 350 | 24.1 | 86.2 | 101 | 227 | 468 | | 225 | 1.01 | 17.3 | 190 | 400 | 190 | 4.47 | 136 | 50.4 | 50.8 | 49.9 |
| MCD-PE | ViT | 112 | 143 | 161 | 1.55 | 108 | 0.05 | 304 | 13.8 | 76.0 | 90.4 | 238 | 487 | | 230 | 0.86 | 18.1 | 195 | 409 | 198 | 3.69 | 136 | 52.0 | 52.4 | 50.9 |
| MCD-EE | ViT | 113 | 144 | 161 | 1.55 | 108 | 0.05 | 304 | 13.8 | 77.2 | 89.9 | 236 | 485 | | 228 | 0.86 | 17.7 | 194 | 407 | 197 | 3.69 | 135 | 51.7 | 51.9 | 50.6 |
| MCD-MI | ViT | 112 | 145 | 161 | 1.55 | 108 | 0.06 | 316 | 13.9 | 76.0 | 93.3 | 244 | 494 | | 234 | 0.87 | 18.9 | 197 | 411 | 200 | 3.72 | 136 | 52.5 | 53.0 | 51.4 |
| MAHA | ViT | 188 | 170 | 360 | 4.71 | 159 | 1.23 | 95.1 | 21.4 | 158 | 83.4 | 219 | 470 | | 222 | 1.48 | 16.0 | 185 | 410 | 185 | 5.21 | 124 | 48.8 | 49.2 | 50.0 |

shifts). Even on the test data they have been proposed on, all three struggle to outperform simple baselines.

These findings indicate a pressing need to evaluate newly proposed CSFs for failure detection in a wide variety of data sets and distribution shifts in order to draw general methodological conclusions.

**Prevalent OoD-D methods are only relevant in a narrow range of distribution shifts.** The proposed evaluation protocol allows, for the first time, to study the relevance of the predominant OoD-D methods in a realistic range of distribution shifts. While for non-semantic new class shifts ("far OoD"), prevalent methods from OoD-D (MLS, MCD-MLS, MAHA) show the best performance across both classifiers, their superiority vanishes already on semantic new class shifts (only ViT-based MAHA on SVHN shows best performance). On the broad range of more nuanced (and arguable more realistic) covariate shifts, however, OoD-D methods are widely outperformed by softmax baselines.

This finding points out an interesting future research direction of developing CSFs that are able to detect failures across the entire range of distribution shifts.

**AURC is able to resolve previous obscurities between classifier robustness and CSF performance.** The results of ConfidNet provide a vivid example of the relevance of assessing classifier performance and confidence ranking in a single score when evaluating CSFs. The original publication reports superior results on CIFAR-10 and CIFAR-100 compared to the MCD-MSR baseline as measured by the MisD metric $AUROC_f$. These results are confirmed in Table 9, but we observe a beneficial effect of MCD on classifier training that leads to improved accuracy (see Table 8). This poses the question: Which of the two methods (ConfidNet or MCD-MSR) will eventually lead to fewer silent failures of the classifier? One directly aids the classifier to produce fewer failures and the other seems better at detecting the existing ones (at least on its test set with potentially more easily preventable failures)? AURC naturally answers this question by expressing the two effects in a single score that directly

Table 2: **Rounding errors occurring during the softmax operation negatively affect the ranking performance of MSR.** The table shows, for different floating point precisions, error rates at which rounding errors occur, affected ranking metrics AURC and $\text{AUROC}_f$, as well as Accuracy for i.i.d. test set performance of MSR. The default setting of 32-bit precision leads to substantial ranking performance drops on the ViT classifier. Note that not all rounding errors necessarily decrease ranking performance, especially in high-accuracy settings where failure cases are rare.

| | | Round-to-one error rate $*100 \downarrow$ | | | AURC $*10^3 \downarrow$ | | | $\text{AUROC}_f * 100 \uparrow$ | | | Accuracy $*100 \uparrow$ |
|---|---|---|---|---|---|---|---|---|---|---|---|
| | | 16bit | 32bit | 64bit | 16bit | 32bit | 64bit | 16bit | 32bit | 64bit | |
| CNN | iWildCam | 47.30 | 1.802 | 0.000 | 82.22 | 69.20 | 69.00 | 85.80 | 87.50 | 87.50 | 76.01 |
| | BREEDS | 35.25 | 4.268 | 0.003 | 18.81 | 12.89 | 12.84 | 89.89 | 92.22 | 92.22 | 90.72 |
| | CAMELYON | 5.365 | 0.001 | 0.000 | 10.25 | 10.12 | 10.12 | 89.18 | 89.21 | 89.21 | 93.99 |
| | CIFAR-100 | 22.75 | 2.264 | 0.001 | 87.22 | 77.43 | 77.39 | 86.38 | 87.29 | 87.29 | 73.26 |
| | CIFAR-10 | 41.54 | 1.465 | 0.000 | 8.346 | 5.620 | 5.617 | 91.98 | 93.73 | 93.73 | 94.35 |
| | SVHN | 41.76 | 17.29 | 0.001 | 8.074 | 4.902 | 4.850 | 89.59 | 92.81 | 92.87 | 96.09 |
| ViT | iWildCam | 44.41 | 14.91 | 0.000 | 221.6 | 177.8 | 177.0 | 75.97 | 80.35 | 80.38 | 62.12 |
| | BREEDS | 80.19 | 52.59 | 0.423 | 11.43 | 4.559 | 1.893 | 72.65 | 88.88 | 94.35 | 97.92 |
| | CAMELYON | 82.03 | 14.52 | 0.000 | 4.661 | 1.007 | 1.007 | 88.59 | 96.42 | 96.42 | 97.95 |
| | CIFAR-100 | 68.65 | 30.27 | 0.000 | 36.27 | 14.95 | 14.23 | 79.29 | 90.10 | 90.29 | 91.62 |
| | CIFAR-10 | 92.16 | 81.79 | 1.883 | 7.614 | 3.480 | 0.950 | 69.30 | 85.85 | 94.90 | 98.76 |
| | SVHN | 69.02 | 47.17 | 0.305 | 16.94 | 8.757 | 5.475 | 68.75 | 83.55 | 88.14 | 97.30 |

relates to the overarching goal of preventing silent failures. This reveals that the MCD-MSR baseline is in fact superior to ConfidNet on the i.i.d. test sets of both CIFAR-10 and CIFAR-100.

**ViT outperforms the CNN classifier on most data sets.** Figure 8 shows a comparative analysis between ViT and CNN classifier across several metrics. As for AURC, ViT outperforms CNN on all datasets except iWildCam, indicating that the domain gap of ImageNet-pretrained representations might be too large for this task. This is an interesting observation, given that CAMELYON featuring images from the biomedical domain could intuitively represent a larger domain gap. Further looking at Accuracy and $\text{AUROC}_f$ performance, we see that performance gains clearly stem from improved classifier accuracy[5], but the CSF ranking performance is on par for ViT and CNN (although the failure detection task might be harder for ViT given fewer detectable failures compared to CNN).

**Different types of uncertainty are empirically not distinguishable.** Considering the associations made in the literature between uncertainty measures and specific types of uncertainty (see Appendix I), we are interested in the extent to which such relations can be confirmed by empirical evidence from our experiments. As an example, we would expect mutual information (MCD-MI) to perform well on new class shifts where model uncertainty should be high and expected entropy (MCD-EE) to perform well on i.i.d. cases where inherent uncertainty in the data (seen during training) is considered the prevalent type of uncertainty. Although, as expected, MCD-EE performs generally better than MCD-MI on i.i.d. test sets, the reverse behavior can not be observed for distribution shifts. Therefore, no clear distinction can be made between *aleatoric* and *epistemic* uncertainty based on the expected benefits of the associated uncertainty measures. Furthermore, no general advantages of entropy-based uncertainty measures over the simple MCD-MSR baseline are observed.

**CSFs beyond Maximum Softmax Response yield well-calibrated scores.** We advocate for a clear purpose statement in research related to confidence scoring, which for most scenarios implies a separation of the tasks of confidence calibration and confidence ranking (see Section 2). Nevertheless, to demonstrate the relevance of our holistic perspective, we extend FD-Shifts to assess the calibration error, a measure previously exclusively applied to softmax outputs, of all considered CSFs. Platt scaling is used to calibrate CSFs with a natural output range beyond $[0, 1]$ (Platt, 1999). Calibration errors of CSFs are reported in Table 10, indicating that currently neglected CSFs beyond MSR provide competitive calibration (e.g. MCD-PE on CNN or MAHA on ViT) and thus constitute appropriate confidence scores to be interpreted directly by the user.

---

[5]Our CNN results are not representative for state-of-the-art CNN performance, since we employ small models such as VGG-13 or ResNet-50, see Appendix E.2

This observation points out a potential research direction, where, analogously to the quest for CSFs that outperform softmax baselines in confidence ranking, it might be possible to identify CSFs that yield better calibration compared to softmax outputs across a wide range of distribution shifts.

**The Maximum Softmax Response baseline is disadvantaged by numerical errors in the standard setting.** Running inference of our empirical study yields terabytes of output data. When tempted to save disk space by storing logits as 16-bit precision floats instead of 32-bit precision, we found the confidence ranking performance of MSR baselines to drop substantially (reduced AURC and $AUROC_f$ scores). This effect is caused by a numerical error, where high logit scores are rounded to 1 during the softmax operation, thereby losing the ranking information between rounded scores. Surprisingly, when returning to 32-bit precision, we found the rate at which rounding errors occur to still be substantial, especially on the ViT classifier (which has higher accuracy and confidence scores compared to the CNN). Table 2 shows error rates as well as affected metrics for different floating point precisions. *Crucially, confidence ranking on ViT classifiers is still affected by rounding errors even in the default 32-bit precision setting* (effects for the CNN are marginal as seen in AURC scores), see for instance $AUROC_f$ drops of $9\%$ on CIFAR-10 and $5.47\%$ on BREEDS (i.e. ImageNet data). This finding has far-reaching implications affecting any ViT-based MSR baseline used for confidence ranking tasks (including current OoD-D literature).

We recommend either casting logits to 64-bit precision (as performed for our study) or performing a temperature scaling prior to the softmax operation in order to minimize the rounding errors.

**Further results.** Despite its relevance for application, the often final step of failure detection, i.e. the definition of a decision threshold on the confidence score, is often neglected in research. In Appendix D we present an approach that does not require the calibration of scores and analyze its reliability under distribution shift. In addition, Appendix G features Accuracy and $AUROC_f$ results for all experiments. For a qualitative study of failure cases, see Appendix H.

## 5 CONCLUSION & TAKE-AWAYS

This work does not propose a novel method, metric, or data set. Instead, following the calls for a more rigorous understanding of existing methods (Lipton and Steinhardt, 2018; Sculley et al., 2018) and evaluation pitfalls (Reinke et al., 2021), its relevance comes from providing compelling theoretical and empirical evidence that a review of current evaluation practices is necessary for all research aiming to detect classification failures. Our results demonstrate vividly that the need for reflection in this field outweighs the need for novelty: *None of the prevalent methods proposed in the literature is able to outperform a softmax baseline across a range of realistic failure sources.*

Therefore, our take-home messages are:

1. Research on confidence scoring (including MisD, OoD-D, PUQ, SC) should come with a clearly defined use case and employ a meaningful evaluation protocol that directly reflects this purpose.

2. If the stated purpose is to detect failures of a classifier, evaluation needs to take into account potential effects on the classifier performance. We recommend AURC as the primary metric, as it combines the two aspects in a single score.

3. Analogously to failure prevention ("robustness"), evaluation of failure detection should include a realistic and nuanced set of distribution shifts covering potential failure sources.

4. Comprehensive evaluation in failure detection requires comparing all relevant solutions towards the same goal including methods from previously separated fields.

5. The inconsistency of our results across data sets indicates the need to evaluate failure detection on a variety of diverse data sets.

6. Logits should be cast to 64-bit precision or temperature-scaled prior to the softmax operation for any ranking-related tasks to avoid subpar softmax baselines.

7. Calibration of confidence scoring functions beyond softmax outputs should be considered as an independent task.

8. Our open-source framework features implementations of baselines, metrics, and data sets that allow researchers to perform meaningful benchmarking of confidence scoring functions.

ACKNOWLEDGEMENTS

This work was funded by Helmholtz Imaging (HI), a platform of the Helmholtz Incubator on Information and Data Science. We thank David Zimmerer and Fabian Isensee for insightful discussions and feedback.

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

## A    Formulation of Failure Sources

In general, we distinguish three sources of error that can cause image classification systems to output false predictions (see Figure 2 for exemplary visualizations). **Inlier Misclassifications**: This type of failure source is defined by occurring on cases that are sampled i.i.d. with respect to the training distribution and is commonly addressed by work in MisD and SC. Possible reasons for occurrence are missing evidence in the image related to the ground truth category, poor model fitting, or data variations that are not considered as distribution shifts yet in a specific use case. **Covariate Shift**: Images subject to a covariate shift (Quionero-Candela et al., 2009) can still be assigned to one of the training categories. Various examples are investigated by recent robustness benchmarks: *Image corruptions shift* (Hendrycks and Dietterich, 2019), *domain shift* such as medical images from different scanners and clinical sites or satellite images from different seasons (Koh et al., 2021), or *subpopulation shift*[6], where unseen semantic variations of the training categories occur such as unseen breeds of an animal category (Santurkar et al., 2021). We summarize *domain shift* and *subpopulation shift* under the term *sub-class shift*. To the best of our knowledge, these relevant *sub-class shifts* have not been studied in the context of confidence scoring before. **New-class Shift**: Images subject to new a class shift can not be assigned to any of the training categories. This type of failure is commonly addressed by work in OoD detection. We follow Ahmed and Courville (2020) in further distinguishing semantic (only foreground object is subject to semantic variations, e.g. previously unseen classes, a.k.a "near OoD" (Winkens et al., 2020)) and non-semantic new-class shifts (context changes, e.g. images from a new data set and classification task, a.k.a "far OoD").

## B    Task Formulations addressing Failure Detection

This section provides a technical exposition of current evaluation protocols for all task formulations stating to address the goal of preventing failures of a related classifier by means of CSFs. Figure 3 gives an overview of relevant metrics in this context.

### B.1    Classification

To set the context we describe the standard evaluation protocol of a classification task, which is denoted in the notation with $y_{cl}$ for the class label, distinguishing it from the failure detection label $y_f$. Given a data set $\{(x_i, y_{cl,i})\}_{i=1}^{N}$ of size $N$ with $(x_i, y_{cl,i})$ independent samples from $\mathcal{X} \times \mathcal{Y}$, and including the discrete class label $y_{cl} \in \mathcal{Y} = \{0, 1, 2, ..., n_{classes} - 1\}$. On this data set a classifier $m(\cdot, w) : \mathcal{X} \rightarrow \mathcal{Y}$ maps from the input images $x$ to the predicted labels $\hat{y}$ with model parameters $w$. $\hat{y}_{m,i}$ is the classification decision obtained as the maximum class probability from the classification model's probability output vector $P_m(y_{cl}|x, w) \in [0, 1]^{n_{classes}}$:

$$\hat{y}_m = \arg\max_{c \in \mathcal{Y}} P_m(y_{cl} = c|x, w), \tag{3}$$

with $c \in \{0, ...n_{classes} - 1\}$ for the respective class. According to this notation, $\arg\max_{c \in Y} P_m(y = c|x_i, w)$. The performance of such a classifier is in multi-class setups often evaluated via accuracy,

$$\text{Accuracy} = \frac{1}{N} \sum_{i}^{N} \mathbb{I}(y_{cl,i} = \hat{y}_{m,i}), \tag{4}$$

where $\mathbb{I}(\text{Event})$ is defined as indicator function:

$$\mathbb{I}(\text{Event}) := \begin{cases} 1, & \text{if the event Event occurs} \\ 0, & \text{otherwise.} \end{cases} \tag{5}$$

Another way is to evaluate classes separately by computing the binary label $y_c \in [0, 1]$ as

$$y_c = \mathbb{I}(y_{cl} = c). \tag{6}$$

---

[6]The term *subpopulation shift* has a different meaning in Koh et al. (2021), where it describes variations of category *frequencies* as opposed to unseen variations

Required for generic assessment of CSFs in the context of failure detection

| Task Formulation | Metric | Classifier Performance | A) Confidence Ranking | B) Confidence Calibration |
|---|---|---|---|---|
| Class. | Accuracy | Considered | Not Considered | Not Considered |
| Class. | AUROC | Considered | Not Considered | Not Considered |
| Class. | AP | Considered | Not Considered | Not Considered |
| Class. | Sens./Prec./.. | Considered | Not Considered | Not Considered |
| MisD | $AUROC_f$ | Not Considered | Considered | Not Considered |
| MisD | $AP_f$ | Not Considered | Considered | Not Considered |
| OoD | $AUROC_{Out}$ | Not Considered | Considered | Not Considered |
| SC | Risk / Coverage | Considered | Considered | Not Considered |
| SC | e-AURC | Not Considered | Considered | Not Considered |
| (SC) | **AURC** | Considered | Considered | Not Considered |
| Calibration | ECE | Not Considered | Not Considered | Considered |
| PUQ | NLL | Does not evaluate CSF, but Predicted class probabilities | Does not evaluate CSF, but Predicted class probabilities | Does not evaluate CSF, but Predicted class probabilities |
| PUQ | Brier-Score | Does not evaluate CSF, but Predicted class probabilities | Does not evaluate CSF, but Predicted class probabilities | Does not evaluate CSF, but Predicted class probabilities |

Legend: Considered / Not Considered / Does not evaluate CSF, but Predicted class probabilities

Figure 3: **Overview of Evaluation Metrics in the Context of Failure Detection.** The three performance aspects related to confidence scoring are classification performance, confidence ranking, and confidence calibration. As argued in this work, the former two are required for evaluating CSFs in the context of failure detection, while confidence calibration usually constitutes a separate task and use case (see Appendix C). Appendix B provides a definition of all metrics as well as their relations and showcases how we naturally converged to considering AURC as the primary metric for evaluation in failure detection.

Following this notation $y_{c,i} = \mathbb{I}(y_{cl,i} = c)$ is the binary class label for sample $i$. Subsequently, the confusion matrix can be computed counting the four possible evaluation outcomes per case and class $c$:

$$\text{TP}_{cl}(\theta, c) = \sum_i^N y_{c,i} \cdot \mathbb{I}(P_m(y_{cl} = c|x_i, w) \geq \theta) \tag{7}$$

$$\text{FP}_{cl}(\theta, c) = \sum_i^N (1 - y_{c,i}) \cdot \mathbb{I}(P_m(y_{cl} = c|x_i, w) \geq \theta) \tag{8}$$

$$\text{TN}_{cl}(\theta, c) = \sum_i^N (1 - y_{c,i}) \cdot \mathbb{I}(P_m(y_{cl} = c|x_i, w) < \theta) \tag{9}$$

$$\text{FN}_{cl}(\theta, c) = \sum_i^N y_{c,i} \cdot \mathbb{I}(P_m(y_{cl} = c|x_i, w) < \theta). \tag{10}$$

These cardinalities are defined depending on the cut-off $\theta$ on the provided predicted class probabilities (PCP). Subsequently, various counting metrics can be applied, for instance Sensitivity (also called recall or true positive rate), False Positive Rate (FPR, or 1 - Specificity), or Precision per class $c$:

$$\text{Sensitivity}_{cl}(\theta, c) = \frac{\text{TP}_{cl}(\theta, c)}{\text{TP}_{cl}(\theta, c) + \text{FN}_{cl}(\theta, c)} \tag{11}$$

$$\text{FPR}_{cl}(\theta, c) = \frac{\text{FP}_{cl}(\theta, c)}{\text{TN}_{cl}(\theta, c) + \text{FP}_{cl}(\theta, c)} \tag{12}$$

$$\text{Precision}_{\text{cl}}(\theta, c) = \frac{\text{TP}_{\text{cl}}(\theta, c)}{\text{TP}_{\text{cl}}(\theta, c) + \text{FP}_{\text{cl}}(\theta, c)}. \tag{13}$$

Next to evaluating these metrics on a certain cut-off $\theta$ on PCPs, often multi-threshold metrics are employed, which scan over cut-offs from all PCP values present in the data set to obtain ROC-curves (Sensitivity plotted over FPR) or Precision-Recall-Curves (PRC, Precision plotted over Sensitivity). Model performance in the form of a single score is extracted via computing the respective area under the curve, i.e. the AUROC for ROC-curves. Here the multi-threshold list $\{\theta_t\}_{t=0}^{T}$ of length $T$ are the cut-off values obtained as the unique values of the descending ranking of all CSF values. The class-wise AUROC values can then be computed as follows:

$$\begin{aligned} \text{AUROC}_{\text{cl}}(c) = \sum_{t=1}^{T} \frac{1}{2} &(\text{FPR}_{\text{cl}}(\theta_t, c) - \text{FPR}_{\text{cl}}(\theta_{(t-1)}, c)) \\ &\cdot (\text{Sensitivity}_{\text{cl}}(\theta_t, c) + \text{Sensitivity}_{\text{cl}}(\theta_{(t-1)}, c)). \end{aligned} \tag{14}$$

The AUPRC for PRC-curves is defined similarly, and commonly approximated by a average precision (AP) score due to interpolation issues:

$$AP_{\text{cl}}(c) = \sum_{t=1}^{T} (\text{Sensitivity}_{\text{cl}}(\theta_t, c) - \text{Sensitivity}_{\text{cl}}(\theta_{(t-1)}, c)) \cdot \text{Precision}_{\text{cl}}(\theta_t, c). \tag{15}$$

Both areas under the curves can be interpreted as *ranking metrics*, i.e. they require cases of the class $c$ to be separated from cases of other classes based on a ranking of PCP values.

## B.2   FAILURE DETECTION

Concrete application of CSFs for the task of detecting failures in order to prevent incorrect predictions of a classifier are formulated in Equation 1 in Section 3. Based on the failure label in Equation 2, which is 1 for classification failure and 0 for success, a confusion matrix can be determined for a cutoff value $\tau$ as follows:

$$\text{TP}_{\text{f}}(\tau) = \sum_{i}^{N} (1 - y_{\text{f},i}) \cdot \mathbb{I}(g(x_i) \geq \tau) \tag{16}$$

$$\text{FP}_{\text{f}}(\tau) = \sum_{i}^{N} y_{\text{f},i} \cdot \mathbb{I}(g(x_i) \geq \tau) \tag{17}$$

$$\text{TN}_{\text{f}}(\tau) = \sum_{i}^{N} y_{\text{f},i} \cdot \mathbb{I}(g(x_i) < \tau) \tag{18}$$

$$\text{FN}_{\text{f}}(\tau) = \sum_{i}^{N} (1 - y_{\text{f},i}) \cdot \mathbb{I}(g(x_i) < \tau). \tag{19}$$

Analogous to the evaluation of the classification performance, one can compute different counting metrics for the confidence ranking task:

$$\text{Sensitivity}_{\text{f}}(\tau) = \frac{\text{TP}_{\text{f}}(\tau)}{\text{TP}_{\text{f}}(\tau) + \text{FN}_{\text{f}}(\tau)} \tag{20}$$

$$\text{FPR}_{\text{f}}(\tau) = \frac{\text{FP}_{\text{f}}(\tau)}{\text{TN}_{\text{f}}(\tau) + \text{FP}_{\text{f}}(\tau)} \tag{21}$$

$$\text{Precision}_{\text{f}}(\tau) = \frac{\text{TP}_{\text{f}}(\tau)}{\text{TP}_{\text{f}}(\tau) + \text{FP}_{\text{f}}(\tau)}. \tag{22}$$

### B.2.1 Misclassification Detection

This evaluation protocol for CSFs directly sticks to the binary classification task between correct and incorrect predictions defined above, and uses ranking metrics analogous to the area under the curves defined for classification evaluation based on a multi-threshold list $\{\tau_t\}_{t=0}^{T}$ of length $T$, which are the cut-off values obtained as the unique values of *ascending* ranking of confidence scores. Most commonly used are the failure detection AUROC:

$$
\begin{aligned}
\text{AUROC}_f &= \sum_{t=1}^{T} (\text{FPR}_\text{f}(\tau_t) - \text{FPR}_\text{f}(\tau_{(t-1)})) \cdot (\text{Sensitivity}_\text{f}(\tau_t) + \text{Sensitivity}_\text{f}(\tau_{(t-1)}))/2 \\
&= \sum_{t=1}^{T} \frac{\sum_i^N y_{\text{f},i} \cdot (\mathbb{I}(g(x_i) \geq \tau_t) - \mathbb{I}(g(x_i) \geq \tau_{(t-1)}))}{\sum_i^N y_{\text{f},i}} \\
&\quad \cdot \frac{\sum_i^N (1 - y_{\text{f},i}) \cdot (\mathbb{I}(g(x_i) \geq \tau_t) + \mathbb{I}(g(x_i) \geq \tau_{(t-1)}))}{2 \cdot \sum_i^N (1 - y_{\text{f},i})},
\end{aligned}
\tag{23}
$$

and the failure detection AUPRC or AP-score:

$$
\begin{aligned}
\text{AP}_f &= \sum_{t=1}^{T} (\text{Sensitivity}_\text{f}(\tau_t) - \text{Sensitivity}_\text{f}(\tau_{t-1})) \cdot \text{Precision}_\text{f}(\tau_t) \\
&= \sum_{t=1}^{T} \frac{\sum_i^N (1 - y_{\text{f},i}) \cdot (\mathbb{I}(g(x_i) \geq \tau_t) - \mathbb{I}(g(x_i) \geq \tau_{(t-1)}))}{\sum_i^N (1 - y_{\text{f},i})} \cdot \frac{\sum_i^N (1 - y_{\text{f},i}) \cdot \mathbb{I}(g(x_i) \geq \tau_t)}{\sum_i^N \mathbb{I}(g(x_i) \geq \tau_t)}.
\end{aligned}
\tag{24}
$$

The data set in these tasks is often biased towards many correct predictions and few incorrect predictions since one does not usually think about preventing failures if the classification performance is very poor in the first place. Consequently, this biases the failure detection AP-score resulting in higher scores. The reverse form defining errors as positive samples, which is not biased in the aforementioned way, is also evaluated:

$$
\text{AP}_{\text{f},err} = \sum_{t=1}^{T} \frac{\sum_i^N y_{\text{f},i} \cdot (\mathbb{I}(g(x_i) < \tau_t) - \mathbb{I}(g(x_i) < \tau_{(t-1)}))}{\sum_i^N y_{\text{f},i}} \cdot \frac{\sum_i^N y_{\text{f},i} \cdot \mathbb{I}(g(x_i) < \tau_t)}{\sum_i^N \mathbb{I}(g(x_i) < \tau_t)}.
\tag{25}
$$

As described in Section 2, this protocol comes with the pitfall of not considering the classifier performance, thus potential effects of CSFs on the classifier performance caused by joined training go unnoticed (for a visualization of this pitfall see Figure 4). This effect can be nailed down to the $(1 - y_{\text{f},i})$ factor of Equation 24: The precision score (second factor of the product) only gets a weight for correct predictions $y_\text{f} = 0$, so by perfect separation, it is possible to traverse all correct predictions (all "weights") while the precision score is one, and any subsequent incorrect predictions are not considered by the metric. This behaviour prevents meaningful comparison of arbitrary CSFs in practice. For instance, consider a comparison between two very standard CSFS, the confidence scores derived from the maximum softmax response (MSR) of a classifier against scores derived from the softmax mean over Monte Carlo Dropout (MCD) sampling of the same classifier. This comparison can already lead to considerable evaluation bias because the classification decision based on MCD means approach most likely produces a different set of failure cases. What's more, associated metrics are notoriously sensitive to such label biases: While it is well-known, that high scores in failure detection $\text{AUROC}_\text{f}$ can be achieved via a bias towards true negatives (such as background objects in object detection), in failure detection $\text{AP}_{\text{f},err}$ with failure as the positive label is used (Hendrycks and Gimpel, 2017; Corbière et al., 2019), which can be equally hacked by biases towards easy-to-detect true positives. As a consequence, this metric typically gives the highest scores at the beginning of model training, when large amounts of failures are produced by the classifier.

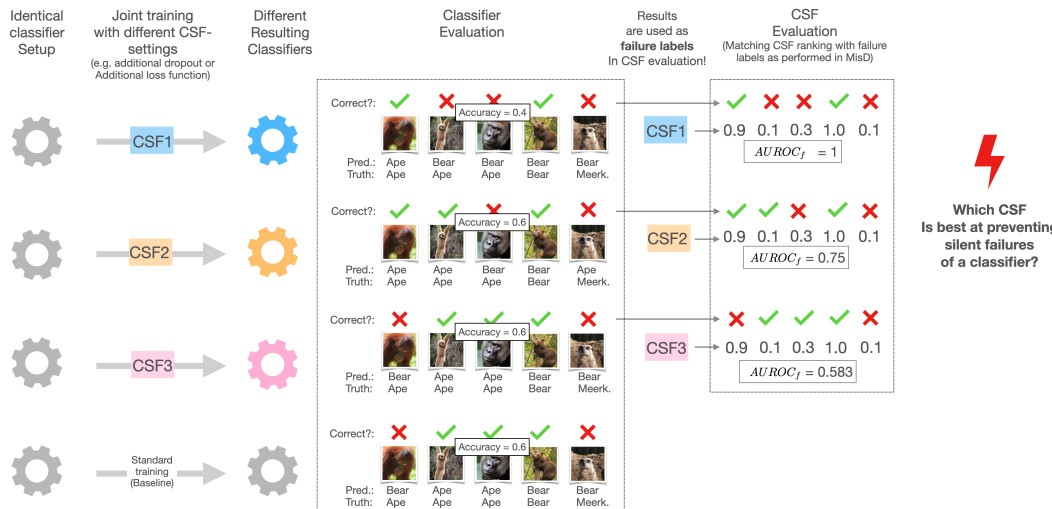

Figure 4: **Visualizing the pitfall of the MisD protocol for failure detection:** This figure demonstrates the need behind Requirement 1 ("Comprehensive evaluation requires a single standardized score that applies to arbitrary CSFs while accounting for their effects on the classifier"). Many CSFs affect the classifier training (e.g. CSFs based on dropout or additional loss functions resulting in distinctive classifiers per CSF. Evaluating the hypothetical task of binary image classification between apes and bears (see "task 1: Failure prevention" in Figure 1), joint training with CSF1 decreases the accuracy of the classifier (compared to training without CSF setting), CSF2 leads to altered predictions with equivalent accuracy, and CSF3 leaves the predictions of the classifier unchanged. Crucially, the results from this evaluation ("failure labels") will be used as reference labels for the CSF evaluation (see "task 2: Failure detection" in Figure 1). For simplicity, we assume that all three CSFs output identical confidence scores. However, ranking metrics such as $\mathrm{AUROC}_f$ (as applied in MisD) yield vastly different results across CSFs because the evaluation is based different sets of reference labels. In this case CSF1 shows the best failure detection performance, but is also the one causing additional failures in the first place. This shows that isolated evaluation of failure detection (as done in MisD) is flawed, because CSF effects on the classifier training are not accounted for. To serve the overarching purpose of preventing silent failures of a classifier ("task1 + task2" in Figure 1), one would need to select a CSF based on both, rankings of Accuracy and $\mathrm{AUROC}_f$. But what is a natural way to weigh the two rankings? The metric proposed in Section 3 directly reflects the purpose of preventing silent failures and can be interpreted as providing a natural weighting between Accuracy and $\mathrm{AUROC}_f$ rankings.

### B.2.2 OUT-OF-DISTRIBUTION DETECTION

In out of distribution detection, the label $y_{\mathrm{out}}$ signals whether a sample is an outlier $y_{\mathrm{out}} = 1$ or an inlier $y_{\mathrm{out}} = 0$. Based on this label the AUROC value is primarily used as a performance metric:

$$
\begin{aligned}
\mathrm{AUROC}_{\mathrm{out}} &= \sum_{t}^{T} (\mathrm{FPR}_{\mathrm{out}}(\tau_t) - \mathrm{FPR}_{\mathrm{out}}(\tau_{(t-1)})) \cdot (\mathrm{Sensitivity}_{\mathrm{out}}(\tau_t) + \mathrm{Sensitivity}_{\mathrm{out}}(\tau_{(t-1)}))/2 \\
&= \sum_{t}^{T} \frac{\sum_{i}^{N} (1 - y_{\mathrm{out},i}) \cdot (\mathbb{I}(g(x_i) \geq \tau_t) - \mathbb{I}(g(x_i) \geq \tau_{(t-1)}))}{\sum_{i}^{N} (1 - y_{\mathrm{out},i})} \cdot \\
&\quad \frac{\sum_{i}^{N} (y_{\mathrm{out},i}) \cdot (\mathbb{I}(g(x_i) \geq \tau_t) + \mathbb{I}(g(x_i) \geq \tau_{(t-1)}))}{2 \cdot \sum_{i}^{N} y_{\mathrm{out},i}}.
\end{aligned}
$$

$$(26)$$

This formulation and the thresholds $\tau_t$ are equivalent to the AUROC used for MisD in Equation 23 except for the different label $y_{\mathrm{out}}$ that is employed (for implications and pitfalls of using this label

see Figure 2). Due to the vast overlap of task formulations FD-Shifts enables a seamless integration of OoD-detection methods for holistic comparison (see Section 3).

### B.2.3 PREDICTIVE UNCERTAINTY ESTIMATION

Common evaluation in PUQ (e.g. Ovadia et al. (2019); Lakshminarayanan et al. (2017) employs proper scoring rules like Negative-Log-Likelihood

$$\text{NLL} = -\frac{1}{N} \sum_i^N \log(P_m(y_{\text{cl}} = y_{\text{cl},i}|x_i, w)), \tag{27}$$

and Brier Score

$$\text{BrierScore} = \frac{1}{N} \sum_i^N \sum_c^{\mathcal{Y}} (P_m(y_{\text{cl}} = c|x_i, w) - y_{c,i})^2. \tag{28}$$

Both directly assess the classifier output and are not applicable to other CSFs. Each metric requires both ranking and calibration of confidence scores which is interpreted as assessing the "general meaningfulness" of scores (see also Figure 6). However, in case there is a clear use case defined requiring either ranking *or* calibration of confidence scores, evaluation with proper scoring rules might dilute this focus and hinder the practical relevance of results.

### B.2.4 SELECTIVE CLASSIFICATION

The idea of equipping a classifier with the option to abstain from certain decisions based on a selection function has been around for a long time (Chow, 1957; El-Yaniv and Wiener, 2010). SC directly evaluates the process of detecting failures of a classifier via CSFs as described by Equation 1. SC essentially tries to optimize the trade-off between achieving low risk and high coverage. Thereby, risk is defined as the error rate of cases remaining after selection:

$$\text{Risk}(\tau) = 1 - \text{Precision}_{\text{f}}(\tau) = \frac{\text{FP}_{\text{f}}(\tau)}{\text{TP}_{\text{f}}(\tau) + \text{FP}_{\text{f}}(\tau)} = \frac{\sum_i^N y_{\text{f},i} \cdot \mathbb{I}(g(x_i) \geq \tau)}{\sum_i^N \mathbb{I}(g(x_i) \geq \tau)} \tag{29}$$

And coverage as the ratio of cases remaining after selection:

$$\text{Coverage}(\tau) = \frac{\text{TP}_{\text{f}}(\tau) + \text{FP}_{\text{f}}(\tau)}{\text{TP}_{\text{f}}(\tau) + \text{FP}_{\text{f}}(\tau) + \text{FN}_{\text{f}}(\tau) + \text{TN}_{\text{f}}(\tau)} = \frac{\sum_i^N \mathbb{I}(g(x_i) \geq \tau)}{N} \tag{30}$$

Typical evaluation is performed for single cut-offs of $\tau$ reporting and comparing resulting risk and coverage scores. One can also calculate the Area under the Risk-Coverage-Curve (AURC):

$$\begin{aligned}
\text{AURC} &= \sum_{t=1}^T (\text{Coverage}(\tau_t) - \text{Coverage}(\tau_{(t-1)})) \cdot (\text{Risk}(\tau_t) + \text{Risk}(\tau_{(t-1)}))/2 \\
&= \sum_{t=1}^T \frac{\sum_i^N (\mathbb{I}(g(x_i) \geq \tau_t) - \mathbb{I}(g(x_i) \geq \tau_{(t-1)}))}{N}. \\
&\quad \left( \frac{\sum_i^N y_{\text{f},i} \cdot \mathbb{I}(g(x_i) \geq \tau_t)}{2 \cdot \sum_i^N \mathbb{I}(g(x_i) \geq \tau_t)} + \frac{\sum_i^N y_{\text{f},i} \cdot \mathbb{I}(g(x_i) \geq \tau_{(t-1)})}{2 \cdot \sum_i^N \mathbb{I}(g(x_i) \geq \tau_{(t-1)})} \right)
\end{aligned} \tag{31}$$

which uses the thresholds $\tau_t$ equivalently to the failure detection AUROC for MisD in Equation 23 as the unique values of the *ascending* ranking of confidence scores. However, AURC is currently not used as the primary metric or part of an established evaluation protocol in SC. Instead, e-AURC has been proposed (Geifman et al., 2019):

$$\text{e-AURC} = \text{AURC} + (1 - \text{Risk}(\tau_0)) \cdot \log(1 - \text{Risk}(\tau_0)) = \text{AURC} - \text{Accuray} \cdot \log(\text{Accuracy}), \tag{32}$$

where it is used that $\text{Risk}(\tau_0) = (1 - \text{Prevalence}_{\text{f}})$ is equal to the Negative Rate. Further, $\text{Prevalence}_{\text{f}}$ is equal to the Accuracy of the base classifier $m$. Therefore e-AURC effectively subtracts the classifier performance aspect from AURC for exclusive focus on evaluating the ranking power of CSFs. This process essentially collapses AURC back to a pure ranking metric such as employed in MisD. However, as laid out in Section 2, evaluating CSFs while being agnostic to the classifier comes with significant pitfalls preventing comprehensive method assessment.

### B.2.5 Unification of Task Formulation - Technical details

In this work, we propose to establish AURC as the primary metric for failure detection, as it fulfills requirements *R1-R3* defined in Section 2. Comparing Equation 31 to Equation 24, a deviation is observed in technical details such as the conservative interpolation of the AP-score versus trapezoidal interpolation in AURC, or the fact that AURC is defined with "reverse precision" in the form of Risk requiring minimization of the score. However, the one crucial conceptual difference is the fact that "weight" is put on the risk score (second factor of the product) in Equation 31 for *all* steps of coverage (first factor). This is in contrast to the $(1 - y_{f,i})$ factor in Equation 24, which ignores steps of incorrect predictions in sensitivity (first factor in the equation), and essentially implies that incorrect predictions of the classifier are penalized irrespective of how perfect the CSF separates these.

We argue that this aspect of evaluation is an essential part of evaluating CSFs in the context of failure detection, because to truly evaluate the property of a CSF to "prevent silent failures of a classifier" we don't only need to check whether CSFs are able to filter existing failures. We also need to check whether they might have caused new failures (or prevented further failures) by altering the training compared to a neutral classifier trained without the CSF, thereby creating their own test set.

In Appendix F we provide a revised implementation of AURC fixing various shortcomings of prior open-source implementations.

Figure 5 visualizes how OoD-D protocols need to be adapted in order to follow the unified task formulation

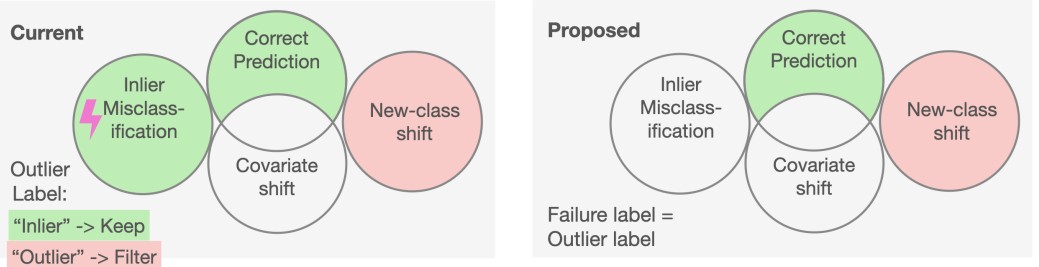

Figure 5: **Modification of the current OoD-Detection protocol for seamless integration into unified evaluation.** Intuitively, data samples labelled as "outlier" can by definition be set to "failure"(or "filter"), because the trained classifier has no possibility to correctly predict unseen classes. However, not all "inlier" are correct classifier decisions. The current OoD-D protocol rewards CSFs for not detecting inlier misclassifications (see Figure 2). On the other hand, penalizing CSFs for not detecting these cases would dilute the desired evaluation focus on new-class shifts. Thus, we propose to remove inlier misclassifications from evaluation when reporting a CSF's performance under new-class shift. While taking those cases out of the *confidence ranking* assessment, we argue that at the same time it is required to fulfil R1, i.e. assess the effect of CSFs on classifier performance. Thus, we propose to report AURC even on new-class shifts instead of $\text{AUROC}_f$, which follows R1 because more inlier failures taken out lead to a higher ratio of OoD cases (failure cases by definition) in the test set and thus a lower accuracy.

## C  Relation between Confidence Ranking and Confidence Calibration

Calibration in the context of CSFs describes the requirement of predicted class scores to match the empirical accuracy of associated cases (e.g. cases with a confidence score of 0.8 are correct $80\%$ of the time), which is useful for use cases like interpretable decision making (e.g. human assessment or a need for interpretable cutoffs like "$risk > 0.8$"), or assessing the applicability of a trained classifier to entire data sets (Guo et al., 2017). Figure 6 shows how confidence calibration and confidence ranking are two independent requirements, i.e. either of the tasks can be solved without solving the other. A CSF can in theory have zero calibration error (perfect calibration) but provide no ranking of

single cases and vice versa, raising the questions: Which requirement do we actually want a CSF to satisfy given a concrete use case? What can we do with calibrated scores in practice if there is no ranking? What are concrete use cases where both tasks are required i.e. a "general meaningfulness" is measured such as by proper scoring rules like negative log-likelihood or Brier Score?

We argue that for all use cases where any sort of selection, cut-off, or separation between cases with lower or higher confidence is performed that does not rely on interpretation of raw confidence values, only confidence ranking and no calibration is required: Given a well-separating CSF (good scores according to ranking metrics), one can define a cut-off value on a validation set according to practical requirements such as "filter only $20\%$ of cases" (coverage guarantee) or "filter such that a maximum of $5\%$ errors remain" (risk guarantee) and select cases according to this cut-off in a subsequent application. Importantly, calibrated scores are not required at any stage in this process. Appendix D demonstrates a realistic example of how to do this under distribution shifts.

While Tomani et al. (2021) studied calibration under a variety of distribution shifts, to the best of our knowledge recent work in the context of neural networks only considers a single CSF: the maximum softmax response (MSR) of the classifier. Applying our holistic perspective, the question arises: If other CSFs beat MSR in confidence ranking tasks, is it not likely that they could also provide better-calibrated scores? To this end, we extend our empirical study and analyze the calibration performance of all studied CSFs under distribution shifts (see Appendix G.

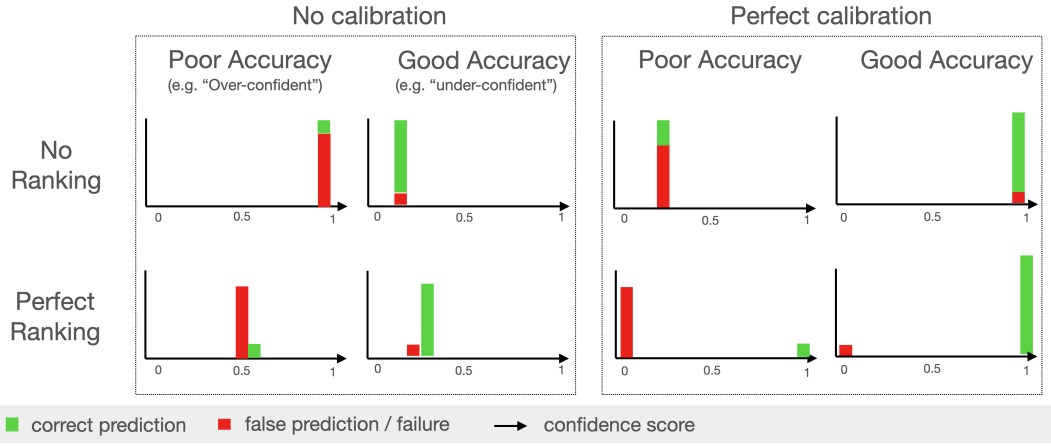

Figure 6: This figure is a rough sketch of the results of a hypothetical classifier to illustrate that the two requirements of calibration and ranking are entirely independent of each other. Note that the confidence scores drawn for the combination of "perfect ranking" and "perfect calibration" (bottom right panels) are not reachable for MSR, because MSR has a lower bound at $1/n_{classes}$.

## D    DEFINING A FINAL DECISION THRESHOLD ON CONFIDENCE SCORES

The idea exists that despite a well-ranked confidence scoring function one still requires calibration of scores in order to set a meaningful decision threshold at the end. In Figure 7 we explore selection under guaranteed risk (SGR) as an alternative decision making tool under distribution shift by exemplary studying ConfidNet under image corruptions on CIFAR-100. Again, comparing the practical applicability to the calibration approach: When a specific application and a potential practitioner observe $0.11$ ECE on the i.i.d. set, what guarantees for reliable decision making can be derived from this score? The results of the SGR study show how given a desired risk $r^*$ the thresholds determined on the validation set provide reliable risk guarantees on the i.i.d. test set. Under image corruption shift, as expected, these guarantees start to break in the order of confidence (here in the statistical sense) parameters $\delta$. While providing risk guarantees under unknown distribution shifts is certainly an open problem, SGR might help practitioners to develop a general feeling for risk excesses under potential shifts in their application domain and to understand how choices of $\delta$ might be able to counteract certain levels of expected shift.

For the experiments on SGR, we use the algorithm proposed by Geifman and El-Yaniv (2017), which computes the threshold $\tau$ that maximizes coverage, while satisfying the following condition:

$$\mathbf{P}_{S_i}\{R(m,g) > r^*\} < \delta, \tag{33}$$

where $\delta$ is a defined confidence parameter, $r^*$ is the desired risk, $R$ the risk and $S_i$ are i.i.d samples from a training or validation set. $m$ is the classifier and $g$ the confidence scoring function defined in Section B

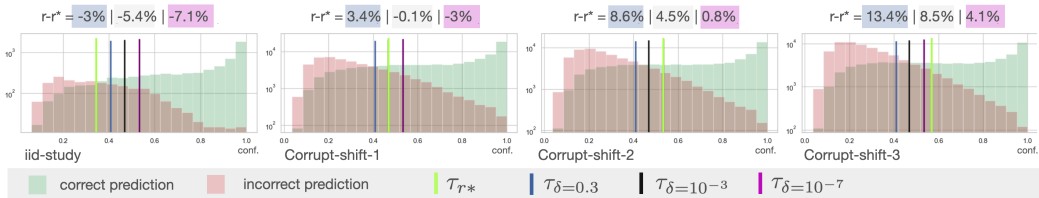

Figure 7: Risk guarantee analysis for ConfidNet on CIFAR-100 under the increasing intensity of image corruptions (logarithmic Y-axis). Thresholds $\tau$ are determined on the validation set via the SGR algorithm from Geifman and El-Yaniv (2017) according to Equation 33 for different confidence parameters $\delta$. The desired risk $r*$ is set to 0.15 and the risk excess $Risk - r*$ under distribution shift is measured. This figure demonstrates how the risk excess between validation set i.i.d. test set is neglectable small, enabling robust decision making without the requirement for calibration. Negative excess values on the i.i.d. study are caused by the confidence $\delta$ on the risk guarantees.

# E    TRAINING DETAILS

## E.1    DATA SET SPLITS

Table 3 shows the number of images per utilized data set and associated splits. We used official splits for all data sets, except that we followed Liu et al. (2019) in splitting a small amount (in our case 1000 images) of the i.i.d. test set for validation and tuning. Because there was no i.i.d. test set for BREEDS-Entity, we held out 10000 training cases for this purpose (1000 of which were again split for validation). The number of images in new-class shifts are sums over the original i.i.d. test splits of SVHN (26032), CIFAR-10 (10000), CIFAR-100 (10000) and TinyImagenet[7] (10000). The model was trained on each case while holding out the test split of the data set. When training on CIFAR-100 super-classes, we discarded the super-class "vehicle 2" because of its obvious semantic overlap with the "vehicle 1" super-class in order to mitigate evaluation noise due to label ambiguities. The sub-class shift test set of the super-classes training was composed by adding up training and testing cases of each held out sub-class per super-class. For the semantic new-class shift on SVHN and iWildCam we sample 40% of classes randomly to be held back and train on the remaining 60% of classes (each of the 5 runs per experiment receives a new sample of held-out classes).

Table 3: Number of images per data set and associated splits.

| Data set | train | val | i.i.d. test | corruptions | sub-class shift | new-class shift |
|---|---|---|---|---|---|---|
| SVHN | 73257 | 1000 | 25032 | - | - | 30000 |
| CIFAR-10 | 50000 | 1000 | 9000 | 750000 | - | 46032 |
| CIFAR-100 | 50000 | 1000 | 9000 | 750000 | - | 46032 |
| CIFAR-100 (super-classes) | 38000 | - | - | - | 11400 | - |
| iWildCam | 129809 | 1000 | 7154 | - | 42791 | - |
| Wilds-Camelyon-17 | 302436 | 1000 | 32560 | - | 85054 | - |
| BREEDS-Entitiy-13 | 157120 | 1000 | 9000 | - | 167592 | - |

---

[7]We used the 32x32 resized data set from `https://github.com/ShiyuLiang/odin-pytorch`

### E.2 Classifier Architectures

For the CNN training, we used a small convolutional network on SVHN (following Corbière et al. (2019)), VGG-13 (Simonyan and Zisserman, 2015) on CIFAR-10/100 (following DeVries and Taylor (2018)), and a ResNet-50 (He et al., 2015) for the three robustness benchmarks. For ViT training we use the ViT-B/16 architecture and scale up all images to 384x384 pixels following Dosovitskiy et al. (2020).

### E.3 Hyperparameters

On SVHN and CIFAR-10/100 we sticked as close as possible to configurations of original publications (Corbière et al., 2019; DeVries and Taylor, 2018; Liu et al., 2019) while at the same time converging to identical classification model configurations. On robustness benchmarks, we stuck to the proposed configurations of respective baseline experiments (Santurkar et al., 2021; Koh et al., 2021). Thus, data augmentation was only used on CIFAR-10/100 (slight rotation, horizontal flip, and cutout following DeVries and Taylor (2018)) and on BREEDS-Entity-13 (horizontal flip and color jitter following Santurkar et al. (2021)). We used a cosine decay schedule without restarts following Loshchilov and Hutter (2017) for all methods and data sets, as we found this schedule to robustly generate high-quality results (see Appendix J for ablations). All classifiers are trained with SGD and momentum 0.9. ConfidNet is trained with Adam, learning rate $10^{-4}$ and finetuned with learning rate $10^{-6}$ following the original configuration. Table 4 shows training parameters that have been modified on each data set. Table 7 shows training parameters used for finetuning the ViT.

Table 4: Training parameters per data set. init-lr: Initial learning rate of the cosine scheduler. wd: L2 weight-decay. DG-CE-epochs: DeepGamblers CrossEntropy pretraining epochs (as proposed in the original implementation (Liu et al., 2019)), ConfidNet-epochs: ConfidNet training epochs (on top of trained classifier) "+" ConfidNet finetuning epochs (for details see Corbière et al. (2019)).

| Data set | init-lr | wd | batch size | epochs | DG-CE-epochs | ConfidNet-epochs |
|---|---|---|---|---|---|---|
| SVHN | $10^{-2}$ | $15 * 10^{-4}$ | 128 | 100 | 50 | 200+20 |
| CIFAR-10 | $10^{-1}$ | $5 * 10^{-4}$ | 128 | 250 | 50 | 200+20 |
| CIFAR-100 | $10^{-1}$ | $5 * 10^{-4}$ | 128 | 250 | 50 | 200+20 |
| iWildCam | $10^{-3}$ | 0 | 16 | 12 | 6 | 5+3 |
| Wilds-Camelyon-17 | $10^{-2}$ | $10^{-2}$ | 32 | 5 | 3 | 3+1 |
| BREEDS-Entity-13 | $10^{-1}$ | $10^{-4}$ | 128 | 300 | 50 | 200+20 |

### E.4 Model Selection

Using the validation set we selected whether or not to use dropout during training for all deterministic confidence scoring functions as well as the best hyperparameter $o$ per DeepGambler-based confidence scoring function. For the latter, we repeated all experiments using $o = [2.2, 3, 6, 10]$. As we found higher $o$ to be beneficial for more complex data sets, we additionally ran $o = [12, 15, 20]$ on CIFAR-100 and $o = 15$ on iWildCam and BREEDS-Entity-13 (additional runs on the last two data sets had to be reduced due to computing resource constraints). Table 5 shows selected $o$ per method and data set, Table 6 shows whether or not dropout has been used for training per method and data set. Table 7 shows whether or not dropout has been used and what learning rate was selected when finetuning the ViT. To select the learning rate we do a single run of training for all learning rates in $[3 * 10^{-2}, 10^{-2}, 3 * 10^{-3}, 10^{-3}, 3 * 10^{-4}, 10^{-4}]$ and chose the lowest AURC on the validation set. If dropout was used to finetune the ViT the dropout rate was 0.1.

Table 5: Selected hyperparameters $o$ based on the validation set for all confidence scoring functions trained with the DeepGamblers objective.

| Method | iWildCam | BREEDS-Entity-13 | Wilds-Camelyon-17 | CIFAR-100 | CIFAR-10 | SVHN |
|---|---|---|---|---|---|---|
| DG-MCD-EE | 10 | 6 | 10 | 20 | 3 | 3 |
| DG-MCD-MSR | 15 | 10 | 10 | 20 | 20 | 6 |
| DG-MCD-PE | 10 | 6 | 10 | 20 | 3 | 3 |
| DG-MSR | 15 | 15 | 2.2 | 12 | 10 | 3 |
| DG-PE | 6 | 15 | 2.2 | 10 | 10 | 3 |
| DG-Res | 6 | 2.2 | 2.2 | 12 | 2.2 | 2.2 |
| DG-Res-MCD | 15 | 3 | 10 | 20 | 2.2 | 2.2 |

Table 6: Whether or not dropout-training has been selected based on the validation set. This selection is only done for deterministic confidence scoring methods (no MCD). "1" denotes dropout training and "0" denotes training without dropout.

| Method | iWildCam | BREEDS-Entity-13 | Wilds-Camelyon-17 | CIFAR-100 | CIFAR-10 | SVHN |
|---|---|---|---|---|---|---|
| ConfidNet | 1 | 0 | 1 | 1 | 1 | 1 |
| DG-MSR | 0 | 0 | 0 | 0 | 0 | 1 |
| DG-PE | 0 | 0 | 0 | 0 | 0 | 1 |
| DG-Res | 0 | 1 | 0 | 0 | 0 | 1 |
| Devries et al. | 1 | 0 | 1 | 1 | 0 | 1 |
| MSR | 0 | 0 | 1 | 0 | 1 | 1 |
| PE | 0 | 0 | 1 | 1 | 1 | 1 |

Table 7: Training parameters and dropout selection per data set and method for ViT. init-lr: Initial learning rate of the cosine scheduler as selected. do: "0" denotes no dropout was used, "1" denotes dropout was used. wd: L2 weight-decay. steps: Number of batches that was trained on.

| Dataset | Method | init-lr | do | wd | batch size | steps |
|---|---|---|---|---|---|---|
| SVHN | MAHA | $3 * 10^{-2}$ | 1 | 0 | 128 | 40000 |
| | MCD | $10^{-2}$ | 1 | | | |
| | MSR | $10^{-2}$ | 0 | | | |
| | PE | $10^{-2}$ | 0 | | | |
| CIFAR-10 | MAHA | $10^{-2}$ | 1 | 0 | 128 | 40000 |
| | MCD | $10^{-2}$ | 1 | | | |
| | MSR | $3 * 10^{-4}$ | 0 | | | |
| | PE | $3 * 10^{-4}$ | 0 | | | |
| CIFAR-100 | MAHA | $10^{-2}$ | 0 | 0 | 512 | 10000 |
| | MCD | $10^{-2}$ | 1 | | | |
| | MSR | $10^{-2}$ | 1 | | | |
| | PE | $10^{-2}$ | 1 | | | |
| CIFAR-100 (super-classes) | MAHA | $3 * 10^{-3}$ | 0 | 0 | 512 | 10000 |
| | MCD | $10^{-3}$ | 1 | | | |
| | MSR | $10^{-3}$ | 1 | | | |
| | PE | $10^{-3}$ | 1 | | | |
| iWildCam | MAHA | $3 * 10^{-3}$ | 0 | 0 | 512 | 40000 |
| | MCD | $3 * 10^{-3}$ | 1 | | | |
| | MSR | $10^{-3}$ | 0 | | | |
| | PE | $10^{-3}$ | 0 | | | |
| Wilds-Camelyon-17 | MAHA | $10^{-3}$ | 0 | 0 | 128 | 40000 |
| | MCD | $3 * 10^{-3}$ | 1 | | | |
| | MSR | $10^{-3}$ | 0 | | | |
| | PE | $10^{-3}$ | 0 | | | |
| BREEDS-Entity-13 | MAHA | $3 * 10^{-3}$ | 0 | 0 | 128 | 40000 |
| | MCD | $10^{-2}$ | 1 | | | |
| | MSR | $10^{-3}$ | 0 | | | |
| | PE | $10^{-3}$ | 0 | | | |

## F   Revised AURC implementation

Our implementation of AURC is based on two implementations we found by Geifman et al.[8] and Corbiere et al.[9]. The two existing implementations have several shortcomings, for instance, in Geifman et al. steps in the RC curve are not defined as new unique values in the sorted list of confidence scores, but instead each data sample, including the ones with equal confidence scores, is considered as leading to an individual classification decision with an associated risk-coverage pair. This effectively leads to a random interpolation of the RC-curve between unique confidence scores adding considerable noise to the result especially in dense singular points such as confidence scores of 0 or 1. A shortcoming in the implementation by Corbiere et al. is that there is no well-defined endpoint of the curve meaning the risk just drops to zero after the last RC-curve step (i.e. the coverage value corresponding to thresholding at the lowest confidence score), which effectively favours methods with higher "lowest coverages". This can make a substantial difference in practice because methods often assign equal confidence scores to more than one case especially at 0 and 1. Thus, analogously to scikit-learn's PRC-curve implementation, we add a final point at zero coverage and the risk remaining the risk of the last RC-curve step.

## G   Additional Results

This section contains additional results for Accuracy (Table 8), $\mathrm{AUROC}_f$ (Table 9, as well as Expected Calibration Error (Table 10). Table 11 shows rankings of confidence scoring functions based on AURC scores (i.e. based on Table 1).

### G.1   Empirical Confirmation of the Importance of Requirements R1-R3

**R1: Comprehensive evaluation requires a single standardized score that applies to arbitrary CSFs while accounting for their effects on the classifier.** The arguments that lead to R1 are stated in Section 2 and visualized in Figure 4. In Table 12 we provide evidence for the importance of this argument: Not following R1 and instead evaluating CSFs with pure rankings metrics like $\mathrm{AUROC}_f$ (as common in MisD) leads to substantially differing rankings of CSFs. This means that the effect of CSFs on the classifier is a crucial factor to consider in practice when ranking CSFs. The finding "AURC is able to resolve previous obscurities between classifier robustness and CSF performance" in Section 4.3 describes a concrete example of how neglecting this factor has led to misleading results in the literature.

Equivalently, we argue that conflating evaluation of failure detection with calibration, as long as the purpose for this dual-purpose assessment is not clearly stated, is a shortcoming of current practices. Empirically showcasing the effects of this conflation on the ranking of CSFs analogously to Table 12 is not possible, because common metrics in PUQ (proper scoring rules) exclusively operate on the predicted class scores and are not compatible with arbitrary CSFs. Instead, this restriction in itself acts as an exclusion criterion for these metrics for the comprehensive evaluation of CSFs.

The experiment shown in Table 12 underlines that fulfilling R1 is essential for meaningful comparison of arbitrary CSFs. Thus, the general findings of our study described in Section 4.3 are all made possible by the proposed protocol following R1 and can be seen as further empirical confirmation of the importance of this requirement.

**R2: Analogously to robustness benchmarks, progress in failure detection requires to evaluate on a nuanced and diverse set of failure sources.** As described in Section 4.3, our study reveals that "none of the evaluated methods from literature beats the simple Maximum Softmax Response baseline across a realistic range of failure sources" and "prevalent OoD-D methods are only relevant in a narrow range of distribution shifts". These findings are a direct result of fulfilling R2 in our evaluation protocol and demonstrate how current protocols (without R2) lead to the proposition of

---

[8]https://github.com/geifmany/uncertainty_ICLR/blob/master/utils/uncertainty_tools.py

[9]https://github.com/valeoai/ConfidNet/blob/master/confidnet/utils/metrics.py

```python
def AURC(residuals, confidence):
    coverages = []
    risks = []
    n = len(residuals)
    idx_sorted = np.argsort(confidence)
    cov = n
    error_sum = sum(residuals[idx_sorted])
    coverages.append(cov/ n),
    risks.append(error_sum / n)
    weights = []
    tmp_weight = 0

    for i in range(0, len(idx_sorted) - 1):

        cov = cov-1
        error_sum = error_sum - residuals[idx_sorted[i]]
        selective_risk = error_sum /(n - 1 - i)
        tmp_weight += 1

        if i == 0 or \
            confidence[idx_sorted[i]] != confidence[idx_sorted[i - 1]]:

             coverages.append(cov / n)
             risks.append(selective_risk)
             weights.append(tmp_weight / n)
             tmp_weight = 0

    if tmp_weight > 0:

        coverages.append(0)
        risks.append(risks[-1])
        weights.append(tmp_weight / n)

    aurc = sum([(risks[i] + risks[i+1]) * 0.5 \
            * weights[i] for i in range(len(weights)) ])

    curve = (coverages, risks)
    return curve, aurc
```

Listing 1: Code for AURC calculation.

Table 8: **Classification performance measured as Accuracy [%] (higher is better).** The color heatmap is normalized per column and classifier (separately for CNN and ViT), while whiter colors depict better scores. "cor" is the average over 5 intensity levels of image corruption shifts. Accuracy scores are averaged over 5 runs per experiment on all data sets except 10 runs on CAMELYON-17-Wilds (as recommended by the authors due to high volatility in results) and 2 runs on BREEDS. Abbreviations: ncs: new-class shift (s for semantic, ns for non-semantic), iid: independent and identically distributed, sub: sub-class shift, cor: image corruptions, c10/100: CIFAR-10/100, ti: TinyImagenet.

| | study | iWildCam iid | sub | s-ncs | BREEDS iid | sub | CAMELYON iid | sub | CIFAR-100 iid | sub | cor | s-ncs c10 | ns-ncs svhn | ti | CIFAR-10 iid | cor | s-ncs c100 | ns-ncs svhn | ti | SVHN iid | s-ncs | ns-ncs c10 | c100 | ti |
|---|---|---|---|---|---|---|---|---|---|---|---|---|---|---|---|---|---|---|---|---|---|---|---|---|
| MSR | CNN | 76.2 | 73.2 | 55.0 | 92.7 | 63.6 | 94.0 | 73.9 | 75.1 | 56.5 | 47.4 | 40.3 | 20.6 | 40.3 | 94.3 | 72.8 | 45.9 | 24.6 | 45.9 | 96.1 | 58.7 | 70.6 | 70.6 | 70.6 |
| MLS | CNN | 76.2 | 73.2 | 55.0 | 92.7 | 63.6 | 94.0 | 73.9 | 73.3 | 55.0 | 44.9 | 39.7 | 20.2 | 39.7 | 94.3 | 72.8 | 45.9 | 24.6 | 45.9 | 96.1 | 58.7 | 70.6 | 70.6 | 70.6 |
| PE | CNN | 76.2 | 73.2 | 55.0 | 92.7 | 63.6 | 94.0 | 73.9 | 73.3 | 55.0 | 44.9 | 39.7 | 20.2 | 39.7 | 94.3 | 72.8 | 45.9 | 24.6 | 45.9 | 96.1 | 58.7 | 70.6 | 70.6 | 70.6 |
| MCD-MSR | CNN | 77.4 | 72.7 | 60.2 | 92.4 | 63.6 | 95.2 | 73.6 | 75.5 | 56.8 | 48.8 | 40.5 | 20.7 | 40.5 | 94.5 | 75.0 | 46.0 | 24.6 | 46.0 | 96.1 | 53.0 | 70.6 | 70.6 | 70.6 |
| MCD-MLS | CNN | 77.4 | 72.7 | 60.2 | 92.4 | 63.6 | 95.1 | 73.0 | 75.5 | 56.8 | 48.8 | 40.5 | 20.7 | 40.5 | 94.5 | 75.1 | 46.0 | 24.6 | 46.0 | 96.1 | 53.0 | 70.6 | 70.6 | 70.6 |
| MCD-PE | CNN | 77.4 | 72.7 | 60.2 | 92.4 | 63.6 | 95.2 | 73.6 | 75.5 | 56.8 | 48.8 | 40.5 | 20.7 | 40.5 | 94.5 | 75.0 | 46.0 | 24.6 | 46.0 | 96.1 | 53.0 | 70.6 | 70.6 | 70.6 |
| MCD-EE | CNN | 77.4 | 72.7 | 60.2 | 92.4 | 63.6 | 95.2 | 73.6 | 75.5 | 56.8 | 48.8 | 40.5 | 20.7 | 40.5 | 94.5 | 75.0 | 46.0 | 24.6 | 46.0 | 96.1 | 53.0 | 70.6 | 70.6 | 70.6 |
| MCD-MI | CNN | 77.4 | 72.7 | 60.2 | 92.4 | 63.6 | 95.2 | 73.6 | 75.5 | 56.8 | 48.8 | 40.5 | 20.7 | 40.5 | 94.5 | 75.0 | 46.0 | 24.6 | 46.0 | 96.1 | 53.0 | 70.6 | 70.6 | 70.6 |
| ConfidNet | CNN | 76.0 | 70.0 | 55.0 | 92.7 | 63.6 | 94.0 | 73.9 | 73.3 | 55.0 | 44.9 | 39.7 | 20.2 | 39.7 | 94.3 | 72.8 | 45.9 | 24.6 | 45.9 | 96.1 | 58.7 | 70.6 | 70.6 | 70.6 |
| DG-MCD-MSR | CNN | 77.0 | 71.2 | 52.4 | 92.9 | 64.4 | 96.5 | 65.2 | 75.4 | 57.0 | 48.9 | 40.4 | 20.7 | 40.4 | 94.3 | 75.2 | 45.9 | 24.6 | 45.9 | 96.1 | 59.8 | 70.6 | 70.6 | 70.6 |
| DG-Res | CNN | 75.5 | 73.9 | 52.5 | 91.5 | 62.5 | 96.3 | 69.5 | 75.2 | 42.0 | 47.5 | 40.4 | 20.6 | 40.4 | 95.0 | 74.2 | 46.1 | 24.7 | 46.1 | 96.1 | 59.7 | 70.6 | 70.6 | 70.6 |
| Devries et al. | CNN | 76.0 | 69.8 | 56.4 | 93.0 | 64.1 | 89.7 | 66.9 | 72.6 | 54.8 | 44.8 | 39.5 | 20.1 | 39.5 | 94.7 | 73.4 | 46.0 | 24.7 | 46.0 | 96.0 | 59.2 | 70.6 | 70.6 | 70.6 |
| | | | | | | | | | | | | | | | | | | | | | | | | |
| MSR | ViT | 72.8 | 74.0 | 49.1 | 97.9 | 75.9 | 99.6 | 83.4 | 91.6 | 79.9 | 77.0 | 45.2 | 24.1 | 45.2 | 98.8 | 93.2 | 47.1 | 25.4 | 47.1 | 97.7 | 57.4 | 71.0 | 71.0 | 71.0 |
| MLS | ViT | 72.8 | 74.0 | 49.1 | 97.9 | 75.9 | 99.6 | 83.4 | 91.6 | 79.9 | 77.0 | 45.2 | 24.1 | 45.2 | 98.8 | 93.2 | 47.1 | 25.4 | 47.1 | 97.7 | 57.4 | 71.0 | 71.0 | 71.0 |
| PE | ViT | 72.8 | 74.0 | 49.1 | 97.9 | 75.9 | 99.6 | 83.4 | 91.6 | 79.9 | 77.0 | 45.2 | 24.1 | 45.2 | 98.8 | 93.2 | 47.1 | 25.4 | 47.1 | 97.7 | 57.4 | 71.0 | 71.0 | 71.0 |
| MCD-MSR | ViT | 68.0 | 59.7 | 61.1 | 97.6 | 74.1 | 99.6 | 60.5 | 90.9 | 76.7 | 72.6 | 45.0 | 23.9 | 45.0 | 98.5 | 89.9 | 47.0 | 25.4 | 47.0 | 97.7 | 57.3 | 71.0 | 71.0 | 71.0 |
| MCD-MLS | ViT | 68.0 | 59.4 | 61.1 | 97.6 | 74.1 | 99.6 | 60.5 | 90.9 | 76.7 | 72.6 | 45.0 | 23.9 | 45.0 | 98.5 | 89.9 | 47.0 | 25.4 | 47.0 | 97.7 | 57.3 | 71.0 | 71.0 | 71.0 |
| MCD-PE | ViT | 68.0 | 59.7 | 61.1 | 97.6 | 74.1 | 99.6 | 60.5 | 90.9 | 76.7 | 72.6 | 45.0 | 23.9 | 45.0 | 98.5 | 89.9 | 47.0 | 25.4 | 47.0 | 97.7 | 57.3 | 71.0 | 71.0 | 71.0 |
| MCD-EE | ViT | 68.0 | 59.7 | 61.1 | 97.6 | 74.1 | 99.6 | 60.5 | 90.9 | 76.7 | 72.6 | 45.0 | 23.9 | 45.0 | 98.5 | 89.9 | 47.0 | 25.4 | 47.0 | 97.7 | 57.3 | 71.0 | 71.0 | 71.0 |
| MCD-MI | ViT | 68.0 | 59.7 | 61.1 | 97.6 | 74.1 | 99.6 | 60.5 | 90.9 | 76.7 | 72.6 | 45.0 | 23.9 | 45.0 | 98.5 | 89.9 | 47.0 | 25.4 | 47.0 | 97.7 | 57.3 | 71.0 | 71.0 | 71.0 |
| MAHA | ViT | 70.9 | 72.0 | 49.1 | 97.9 | 75.9 | 99.6 | 83.4 | 91.6 | 79.9 | 77.0 | 45.2 | 24.1 | 45.2 | 98.8 | 93.2 | 47.1 | 25.4 | 47.1 | 97.7 | 57.4 | 71.0 | 71.0 | 71.0 |

CSFs with a lack of generalization ability that often directly contradicts their purpose statement: to enable the safe application of classifiers by detecting silent failures.

**R3: If there is a defined classifier whose incorrect predictions are to be detected, its respective failure information should be used to assess CSFs w.r.t the stated purpose instead of a surrogate task such as distribution shift detection.** The fact that evaluation w.r.t failure labels, as opposed to OoD-labels, enables testing methods on a broad and realistic range of distribution shifts is the most important argument for R3. This importance is empirically confirmed in the finding "prevalent OoD-D methods are only relevant in a narrow range of distribution shifts" (see Section 4.3), which reveals a clear contradiction of many of these methods' purpose statements and their actual functionality.

Besides the general importance of following R3, we would like to empirically highlight the importance of dismissing inlier misclassifications from CSF evaluation on new-class shifts (the proposed protocol for new-class shifts is visualized in Figure 5). We argue that when assessing *confidence ranking* w.r.t failure detection on new class shifts, penalizing the detection of failures from the i.i.d. set is an undesired behavior of the current OoD-D protocol and subsequently propose to dismiss these cases from ranking (they still affect the classifier performance considered by AURC thus fulfilling R1). Table 13 demonstrates the importance of this modification revealing considerable effects on the resulting CSF rankings (as measured by $\mathrm{AUROC}_f$).

Table 9: **Confidence Ranking results measured as** $\mathrm{AUROC_f}$ [%] **(higher is better).** The color heatmap is normalized per column and classifier (separately for CNN and ViT), while whiter colors depict better scores. "cor" is the average over 5 intensity levels of image corruption shifts. $\mathrm{AUROC_f}$ scores are averaged over 5 runs per experiment on all data sets except 10 runs on CAMELYON-17-Wilds (as recommended by the authors due to high volatility in results) and 2 runs on BREEDS. Abbreviations: ncs: new-class shift (s for semantic, ns for non-semantic), iid: independent and identically distributed, sub: sub-class shift, cor: image corruptions, c10/100: CIFAR-10/100, ti: TinyImagenet

| | study | iWildCam iid | iWildCam sub | iWildCam s-ncs | BREEDS iid | BREEDS sub | CAMELYON iid | CAMELYON sub | CIFAR-100 iid | CIFAR-100 sub | CIFAR-100 cor | CIFAR-100 s-ncs | CIFAR-100 ns-ncs c10 | CIFAR-100 ns-ncs svhn | CIFAR-100 ns-ncs ti | CIFAR-10 iid | CIFAR-10 cor | CIFAR-10 s-ncs | CIFAR-10 ns-ncs c100 | CIFAR-10 ns-ncs svhn | CIFAR-10 ns-ncs ti | SVHN iid | SVHN s-ncs | SVHN ns-ncs c10 | SVHN ns-ncs c100 | SVHN ns-ncs ti |
|---|---|---|---|---|---|---|---|---|---|---|---|---|---|---|---|---|---|---|---|---|---|---|---|---|---|---|
| MSR | CNN | 89.1 | 90.3 | 85.5 | 93.8 | 78.0 | 89.2 | 71.2 | 87.4 | 77.1 | 80.8 | 86.4 | 88.4 | 84.3 | | 93.7 | 86.5 | 91.7 | 95.8 | 96.9 | | 92.9 | 93.9 | 97.5 | 97.3 | 97.4 |
| MLS | CNN | 82.3 | 86.5 | 82.8 | 90.2 | 76.2 | 89.2 | 71.2 | 84.8 | 78.0 | 81.2 | 86.8 | 87.9 | 93.9 | | 92.6 | 86.2 | 92.9 | 97.6 | 98.5 | | 91.5 | 94.3 | 98.6 | 98.3 | 98.4 |
| PE | CNN | 89.0 | 90.2 | 86.1 | 93.8 | 78.2 | 89.2 | 71.2 | 87.4 | 77.9 | 81.9 | 86.0 | 81.4 | 91.0 | | 93.8 | 86.8 | 92.0 | 96.4 | 97.5 | | 92.9 | 94.0 | 97.9 | 97.6 | 97.7 |
| MCD-MSR | CNN | 90.2 | 91.4 | 85.6 | 94.3 | 78.2 | 90.6 | 70.0 | 87.5 | 78.4 | 82.1 | 86.3 | 86.4 | 84.8 | | 94.1 | 86.6 | 91.8 | 94.3 | 95.9 | | 93.0 | 94.6 | 97.9 | 97.7 | 98.0 |
| MCD-MLS | CNN | 81.8 | 88.2 | 84.7 | 90.7 | 76.3 | 90.4 | 69.8 | 84.1 | 78.0 | 80.4 | 86.1 | 88.7 | 87.9 | | 92.3 | 85.8 | 92.9 | 96.2 | 98.0 | | 91.7 | 95.4 | 98.7 | 98.5 | 98.6 |
| MCD-PE | CNN | 89.8 | 90.9 | 86.8 | 94.2 | 78.4 | 90.6 | 70.0 | 85.6 | 78.3 | 81.0 | 86.2 | 87.4 | 87.1 | | 93.7 | 86.6 | 92.4 | 95.0 | 96.9 | | 92.8 | 94.9 | 98.4 | 98.2 | 98.4 |
| MCD-EE | CNN | 89.6 | 90.7 | 86.8 | 94.1 | 78.3 | 90.5 | 70.0 | 85.6 | 78.2 | 80.9 | 85.9 | 86.5 | 86.5 | | 93.6 | 86.8 | 92.5 | 96.2 | 97.5 | | 92.7 | 95.1 | 98.5 | 98.2 | 98.4 |
| MCD-MI | CNN | 89.7 | 89.4 | 76.9 | 93.7 | 77.5 | 89.5 | 66.8 | 84.1 | 77.5 | 79.2 | 85.6 | 88.6 | 87.8 | | 93.3 | 84.5 | 91.6 | 90.4 | 93.7 | | 92.6 | 94.3 | 97.8 | 97.5 | 98.1 |
| ConfidNet | CNN | 70.8 | 77.9 | 86.6 | 93.8 | 77.9 | 95.6 | 73.6 | 88.9 | 78.4 | 84.6 | 86.1 | 86.5 | 91.4 | | 94.2 | 87.8 | 92.2 | 96.3 | 97.1 | | 92.9 | 94.0 | 97.4 | 97.2 | 97.3 |
| DG-MCD-MSR | CNN | 90.0 | 91.1 | 87.6 | 94.3 | 78.6 | 92.4 | 62.8 | 87.9 | 78.4 | 82.0 | 86.2 | 86.7 | 83.6 | | 94.2 | 86.1 | 91.2 | 93.4 | 94.8 | | 93.1 | 94.7 | 97.8 | 97.6 | 97.8 |
| DG-Res | CNN | 80.2 | 84.5 | 85.4 | 92.6 | 77.5 | 92.6 | 65.6 | 82.1 | 74.4 | 77.1 | 84.4 | 90.7 | 89.1 | | 93.4 | 86.1 | 90.9 | 97.6 | 97.8 | | 91.3 | 94.2 | 98.5 | 98.3 | 98.2 |
| Devries et al. | CNN | 81.9 | 86.3 | 81.0 | 91.0 | 75.9 | 83.7 | 63.1 | 84.2 | 76.4 | 78.2 | 84.1 | 74.8 | 80.4 | | 93.7 | 84.0 | 91.8 | 97.8 | 97.5 | | 87.4 | 88.0 | 97.4 | 96.7 | 97.5 |
| MSR | ViT | 86.5 | 89.0 | 86.1 | 94.3 | 78.2 | 98.1 | 78.6 | 90.3 | 81.8 | 87.3 | 91.9 | 93.8 | 89.7 | | 94.9 | 92.8 | 97.2 | 98.8 | 95.3 | | 90.2 | 93.8 | 98.1 | 98.0 | 98.4 |
| MLS | ViT | 81.2 | 84.8 | 86.5 | 89.4 | 74.2 | 97.3 | 78.9 | 84.5 | 76.9 | 84.9 | 94.3 | 95.4 | 91.2 | | 92.8 | 91.9 | 97.5 | 99.5 | 96.0 | | 87.8 | 94.2 | 98.7 | 98.5 | 99.0 |
| PE | ViT | 86.7 | 89.0 | 86.6 | 94.3 | 78.2 | 98.1 | 78.6 | 90.3 | 81.8 | 87.5 | 92.2 | 94.2 | 90.0 | | 94.9 | 92.8 | 97.2 | 98.8 | 95.3 | | 90.2 | 93.8 | 98.2 | 98.1 | 98.4 |
| MCD-MSR | ViT | 84.5 | 87.4 | 84.5 | 95.6 | 79.5 | 98.9 | 64.4 | 91.2 | 83.2 | 86.8 | 91.2 | 89.0 | 92.7 | | 96.0 | 91.8 | 96.1 | 97.6 | 95.8 | | 90.8 | 95.5 | 98.1 | 98.0 | 98.5 |
| MCD-MLS | ViT | 79.5 | 85.9 | 84.4 | 92.4 | 76.0 | 98.0 | 58.9 | 84.7 | 80.3 | 84.9 | 94.0 | 93.7 | 94.9 | | 95.1 | 92.1 | 97.6 | 99.6 | 97.6 | | 88.6 | 96.0 | 98.9 | 98.8 | 99.1 |
| MCD-PE | ViT | 84.7 | 87.8 | 85.3 | 95.6 | 79.6 | 98.9 | 64.4 | 91.1 | 83.3 | 87.1 | 91.9 | 89.9 | 93.4 | | 96.0 | 91.9 | 96.2 | 97.7 | 96.0 | | 90.8 | 95.6 | 98.3 | 98.2 | 98.6 |
| MCD-EE | ViT | 84.4 | 87.6 | 85.3 | 95.6 | 79.4 | 98.9 | 64.3 | 91.1 | 82.8 | 87.2 | 92.3 | 90.5 | 93.7 | | 96.0 | 92.2 | 96.5 | 98.2 | 96.3 | | 90.8 | 95.8 | 98.4 | 98.4 | 98.8 |
| MCD-MI | ViT | 84.6 | 87.5 | 84.8 | 95.6 | 79.4 | 98.8 | 63.6 | 90.9 | 83.2 | 86.1 | 90.3 | 87.8 | 92.3 | | 95.9 | 91.4 | 95.7 | 97.0 | 95.4 | | 90.8 | 95.5 | 98.1 | 97.9 | 98.5 |
| MAHA | ViT | 67.7 | 69.2 | 71.3 | 85.2 | 63.4 | 77.9 | 68.2 | 83.0 | 57.6 | 81.8 | 94.3 | 90.0 | 94.7 | | 90.4 | 86.1 | 98.2 | 97.6 | 98.3 | | 86.4 | 97.3 | 99.3 | 99.2 | 98.9 |

Table 10: **Confidence Calibration Results measured as Expected Calibration Error (lower is better).** To our knowledge, this table shows the most comprehensive study of confidence calibration to date featuring CSFs beyond MSR and including a broad range of distribution shifts. Importantly, only the CSFs with a natural output range beyond [0,1] have undergone Platt-scaling, so the data does not provide a fair comparison of calibration errors between CSFs. Instead, the purpose is to demonstrate that CSFs beyond MSR can be calibrated and could thus be considered an appropriate competition to MSR in future research. The color heatmap is normalized per column and classifier (separately for CNN and ViT), while whiter colors depict better scores. "cor" is the average over 5 intensity levels of image corruption shifts. ECE scores are averaged over 5 runs per experiment on all data sets except 10 runs on CAMELYON-17-Wilds (as recommended by the authors due to high volatility in results) and 2 runs on BREEDS. Abbreviations: ncs: new-class shift (s for semantic, ns for non-semantic), iid: independent and identically distributed, sub: sub-class shift, cor: image corruptions, c10/100: CIFAR-10/100, ti: TinyImagenet

| | | iWildCam | | | BREEDS | | CAMELYON | | CIFAR-100 | | | | | | CIFAR-10 | | | | | SVHN | | | | |
| | study | iid | sub | s-ncs | iid | sub | iid | sub | iid | sub | cor | s-ncs | ns-ncs | | iid | cor | s-ncs | ns-ncs | | iid | s-ncs | ns-ncs | | |
| | ncs-data set | | | | | | | | | | | c10 | svhn | ti | | | c100 | svhn | ti | | | c10 | c100 | ti |
|---|---|---|---|---|---|---|---|---|---|---|---|---|---|---|---|---|---|---|---|---|---|---|---|---|
| MSR | CNN | 0.14 | 0.11 | 0.22 | 0.03 | 0.22 | 0.01 | 0.14 | 0.06 | 0.27 | 0.19 | 0.28 | 0.41 | 0.31 | 0.03 | 0.16 | 0.43 | 0.57 | 0.35 | 0.02 | 0.34 | 0.19 | 0.19 | 0.19 |
| MLS | CNN | 0.07 | 0.10 | 0.10 | 0.01 | 0.15 | 0.01 | 0.15 | 0.01 | 0.02 | 0.11 | 0.21 | 0.34 | 0.15 | 0.01 | 0.07 | 0.34 | 0.41 | 0.22 | 0.01 | 0.02 | 0.13 | 0.13 | 0.13 |
| PE | CNN | 0.12 | 0.10 | 0.10 | 0.03 | 0.17 | 0.01 | 0.15 | 0.09 | 0.09 | 0.16 | 0.23 | 0.42 | 0.16 | 0.02 | 0.11 | 0.36 | 0.42 | 0.20 | 0.02 | 0.12 | 0.12 | 0.12 | 0.12 |
| MCD-MSR | CNN | 0.10 | 0.07 | 0.17 | 0.02 | 0.18 | 0.02 | 0.14 | 0.03 | 0.15 | 0.07 | 0.19 | 0.32 | 0.20 | 0.01 | 0.07 | 0.35 | 0.49 | 0.29 | 0.01 | 0.33 | 0.13 | 0.13 | 0.13 |
| MCD-MLS | CNN | 0.09 | 0.14 | 0.10 | 0.01 | 0.15 | 0.01 | 0.16 | 0.02 | 0.02 | 0.09 | 0.22 | 0.34 | 0.21 | 0.01 | 0.07 | 0.35 | 0.49 | 0.25 | 0.01 | 0.02 | 0.12 | 0.12 | 0.12 |
| MCD-PE | CNN | 0.11 | 0.10 | 0.08 | 0.03 | 0.17 | 0.01 | 0.16 | 0.05 | 0.05 | 0.10 | 0.21 | 0.35 | 0.20 | 0.02 | 0.08 | 0.33 | 0.48 | 0.23 | 0.01 | 0.09 | 0.10 | 0.10 | 0.10 |
| MCD-EE | CNN | 0.12 | 0.10 | 0.08 | 0.03 | 0.17 | 0.01 | 0.16 | 0.06 | 0.06 | 0.11 | 0.21 | 0.37 | 0.22 | 0.02 | 0.08 | 0.32 | 0.40 | 0.18 | 0.02 | 0.09 | 0.09 | 0.10 | 0.10 |
| MCD-MI | CNN | 0.09 | 0.10 | 0.15 | 0.03 | 0.20 | 0.02 | 0.20 | 0.04 | 0.04 | 0.11 | 0.22 | 0.32 | 0.19 | 0.02 | 0.12 | 0.36 | 0.63 | 0.38 | 0.01 | 0.10 | 0.13 | 0.13 | 0.11 |
| ConfidNet | CNN | 0.20 | 0.23 | 0.17 | 0.02 | 0.18 | 0.05 | 0.08 | 0.05 | 0.21 | 0.15 | 0.25 | 0.38 | 0.18 | 0.02 | 0.11 | 0.38 | 0.47 | 0.28 | 0.02 | 0.37 | 0.19 | 0.19 | 0.19 |
| DG-MCD-MSR | CNN | 0.11 | 0.09 | 0.27 | 0.02 | 0.18 | 0.02 | 0.21 | 0.02 | 0.15 | 0.09 | 0.21 | 0.34 | 0.23 | 0.01 | 0.09 | 0.37 | 0.52 | 0.32 | 0.01 | 0.28 | 0.14 | 0.14 | 0.13 |
| DG-Res | CNN | 0.24 | 0.25 | 0.36 | 0.04 | 0.20 | 0.02 | 0.28 | 0.25 | 0.57 | 0.52 | 0.59 | 0.79 | 0.59 | 0.04 | 0.19 | 0.47 | 0.57 | 0.30 | 0.04 | 0.40 | 0.29 | 0.29 | 0.29 |
| Devries et al. | CNN | 0.09 | 0.09 | 0.20 | 0.04 | 0.11 | 0.16 | 0.14 | 0.09 | 0.16 | 0.28 | 0.34 | 0.57 | 0.38 | 0.10 | 0.14 | 0.14 | 0.08 | 0.04 | 0.16 | 0.11 | 0.09 | 0.08 | 0.09 |
| | | | | | | | | | | | | | | | | | | | | | | | | |
| MSR | ViT | 0.19 | 0.17 | 0.37 | 0.02 | 0.21 | 0.00 | 0.14 | 0.06 | 0.15 | 0.17 | 0.43 | 0.59 | 0.45 | 0.01 | 0.06 | 0.48 | 0.69 | 0.49 | 0.02 | 0.40 | 0.25 | 0.25 | 0.24 |
| MLS | ViT | 0.06 | 0.06 | 0.14 | 0.01 | 0.13 | 0.00 | 0.11 | 0.02 | 0.02 | 0.06 | 0.23 | 0.34 | 0.27 | 0.00 | 0.03 | 0.28 | 0.33 | 0.29 | 0.01 | 0.02 | 0.11 | 0.11 | 0.10 |
| PE | ViT | 0.10 | 0.10 | 0.17 | 0.00 | 0.18 | 0.00 | 0.14 | 0.04 | 0.05 | 0.10 | 0.29 | 0.40 | 0.32 | 0.00 | 0.05 | 0.42 | 0.63 | 0.44 | 0.01 | 0.10 | 0.19 | 0.19 | 0.18 |
| MCD-MSR | ViT | 0.20 | 0.22 | 0.27 | 0.01 | 0.17 | 0.00 | 0.37 | 0.04 | 0.12 | 0.14 | 0.37 | 0.56 | 0.35 | 0.01 | 0.06 | 0.43 | 0.63 | 0.44 | 0.01 | 0.36 | 0.21 | 0.21 | 0.20 |
| MCD-MLS | ViT | 0.07 | 0.10 | 0.13 | 0.01 | 0.10 | 0.00 | 0.38 | 0.02 | 0.03 | 0.06 | 0.26 | 0.40 | 0.24 | 0.01 | 0.04 | 0.32 | 0.39 | 0.32 | 0.01 | 0.02 | 0.12 | 0.12 | 0.12 |
| MCD-PE | ViT | 0.10 | 0.12 | 0.15 | 0.01 | 0.16 | 0.00 | 0.37 | 0.04 | 0.06 | 0.09 | 0.27 | 0.44 | 0.24 | 0.00 | 0.06 | 0.38 | 0.59 | 0.39 | 0.01 | 0.11 | 0.14 | 0.15 | 0.13 |
| MCD-EE | ViT | 0.12 | 0.14 | 0.16 | 0.01 | 0.17 | 0.00 | 0.37 | 0.05 | 0.08 | 0.10 | 0.25 | 0.41 | 0.22 | 0.01 | 0.06 | 0.33 | 0.48 | 0.34 | 0.01 | 0.10 | 0.13 | 0.13 | 0.12 |
| MCD-MI | ViT | 0.09 | 0.11 | 0.16 | 0.01 | 0.17 | 0.00 | 0.39 | 0.04 | 0.06 | 0.10 | 0.36 | 0.57 | 0.31 | 0.01 | 0.06 | 0.47 | 0.69 | 0.47 | 0.01 | 0.09 | 0.18 | 0.18 | 0.17 |
| MAHA | ViT | 0.05 | 0.06 | 0.16 | 0.00 | 0.12 | 0.00 | 0.13 | 0.02 | 0.03 | 0.08 | 0.28 | 0.55 | 0.24 | 0.00 | 0.04 | 0.32 | 0.60 | 0.27 | 0.01 | 0.03 | 0.11 | 0.11 | 0.10 |

Table 11: **Failure Detection Results shown as Rankings according to AURC.** While providing results in the form of rankings might facilitate parsing the information quickly, it should be said that rankings hide important information in method assessment. Specifically, the notion to distinguish between "similar performance" and "substantially different" performance is lost. The color heatmap is normalized per column and classifier (separately for CNN and ViT), while whiter colors depict better scores. "cor" is the average over 5 intensity levels of image corruption shifts. Scores are averaged over 5 runs per experiment on all data sets except 10 runs on CAMELYON-17-Wilds (as recommended by the authors due to high volatility in results) and 2 runs on BREEDS. Abbreviations: ncs: new-class shift (s for semantic, ns for non-semantic), iid: independent and identically distributed, sub: sub-class shift, cor: image corruptions, c10/100: CIFAR-10/100, ti: TinyImagenet.

| | study | iWildCam | | | BREEDS | | CAMELYON | | CIFAR-100 | | | | | | | CIFAR-10 | | | | | | SVHN | | | | |
| --- | --- | --- | --- | --- | --- | --- | --- | --- | --- | --- | --- | --- | --- | --- | --- | --- | --- | --- | --- | --- | --- | --- | --- | --- | --- | --- |
| | ncs-data set | iid | sub | s-ncs | iid | sub | iid | sub | iid | sub | cor | s-ncs | c10 | svhn | ti | iid | cor | s-ncs | c100 | svhn | ti | iid | s-ncs | c10 | c100 | ti |
| MSR | CNN | 6 | 4 | 7 | 5 | 6 | 9 | 2 | 3 | 7 | 7 | 3 | 3 | | 10 | 9 | 10 | 7 | 7 | | 8 | 5 | 4 | 10 | 10 | 11 |
| MLS | CNN | 9 | 9 | 11 | 12 | 12 | 9 | 2 | 10 | 9 | 11 | 8 | 6 | | 1 | 12 | 11 | 1 | 2 | | 1 | 10 | 3 | 2 | 2 | 4 |
| PE | CNN | 7 | 5 | 6 | 7 | 5 | 9 | 2 | 8 | 10 | 9 | 9 | 11 | | 3 | 8 | 9 | 6 | 4 | | 5 | 5 | 4 | 7 | 7 | 9 |
| MCD-MSR | CNN | 1 | 1 | 3 | 2 | 2 | 4 | 5 | 2 | 2 | 2 | 1 | 8 | | 9 | 2 | 3 | 7 | 10 | | 10 | 2 | 11 | 6 | 6 | 7 |
| MCD-MLS | CNN | 8 | 8 | 4 | 11 | 11 | 7 | 8 | 9 | 5 | 5 | 7 | 2 | | 5 | 11 | 5 | 1 | 4 | | 2 | 9 | 8 | 1 | 1 | 1 |
| MCD-PE | CNN | 2 | 2 | 1 | 3 | 2 | 4 | 5 | 4 | 2 | 3 | 3 | 5 | | 7 | 6 | 2 | 4 | 9 | | 9 | 4 | 10 | 5 | 5 | 2 |
| MCD-EE | CNN | 4 | 3 | 1 | 4 | 2 | 6 | 5 | 5 | 4 | 4 | 5 | 6 | | 8 | 7 | 1 | 3 | 7 | | 5 | 5 | 9 | 4 | 4 | 3 |
| MCD-MI | CNN | 2 | 6 | 8 | 5 | 7 | 8 | 9 | 7 | 6 | 6 | 5 | 4 | | 6 | 10 | 6 | 10 | 12 | | 12 | 8 | 12 | 9 | 9 | 6 |
| ConfidNet | CNN | 12 | 12 | 5 | 8 | 7 | 3 | 1 | 6 | 8 | 8 | 10 | 10 | | 2 | 4 | 7 | 4 | 4 | | 7 | 3 | 4 | 11 | 11 | 12 |
| DG-MCD-MSR | CNN | 5 | 7 | 9 | 1 | 1 | 2 | 11 | 1 | 1 | 1 | 1 | 9 | | 11 | 3 | 4 | 11 | 11 | | 11 | 1 | 1 | 8 | 8 | 8 |
| DG-Res | CNN | 11 | 10 | 12 | 10 | 10 | 1 | 10 | 11 | 12 | 10 | 11 | 1 | | 4 | 5 | 8 | 12 | 3 | | 3 | 11 | 2 | 3 | 3 | 5 |
| Devries et al. | CNN | 10 | 11 | 10 | 9 | 9 | 12 | 12 | 12 | 11 | 12 | 12 | 12 | | 12 | 1 | 12 | 7 | 1 | | 3 | 12 | 7 | 12 | 12 | 10 |
| | | | | | | | | | | | | | | | | | | | | | | | | | | |
| MSR | ViT | 2 | 2 | 7 | 5 | 1 | 5 | 2 | 5 | 1 | 2 | 6 | 3 | | 9 | 5 | 2 | 3 | 3 | | 8 | 5 | 8 | 9 | 9 | 9 |
| MLS | ViT | 3 | 3 | 8 | 8 | 7 | 8 | 1 | 8 | 7 | 3 | 2 | 1 | | 7 | 8 | 3 | 3 | 2 | | 7 | 8 | 7 | 3 | 4 | 3 |
| PE | ViT | 1 | 1 | 6 | 5 | 1 | 5 | 2 | 5 | 1 | 1 | 4 | 2 | | 8 | 5 | 1 | 3 | 3 | | 8 | 5 | 8 | 7 | 8 | 8 |
| MCD-MSR | ViT | 6 | 6 | 4 | 1 | 3 | 1 | 5 | 1 | 5 | 7 | 8 | 8 | | 5 | 1 | 8 | 8 | 7 | | 4 | 3 | 3 | 6 | 6 | 6 |
| MCD-MLS | ViT | 8 | 8 | 5 | 7 | 8 | 7 | 9 | 9 | 8 | 9 | 3 | 4 | | 2 | 7 | 5 | 2 | 1 | | 2 | 7 | 3 | 2 | 2 | 1 |
| MCD-PE | ViT | 4 | 4 | 1 | 2 | 3 | 1 | 5 | 2 | 3 | 6 | 6 | 7 | | 4 | 1 | 7 | 7 | 6 | | 4 | 1 | 3 | 5 | 5 | 5 |
| MCD-EE | ViT | 6 | 5 | 1 | 2 | 3 | 1 | 5 | 2 | 6 | 5 | 4 | 6 | | 3 | 1 | 6 | 6 | 5 | | 3 | 1 | 2 | 4 | 3 | 4 |
| MCD-MI | ViT | 4 | 6 | 1 | 2 | 3 | 4 | 8 | 4 | 3 | 8 | 9 | 9 | | 6 | 4 | 9 | 9 | 9 | | 6 | 4 | 3 | 7 | 7 | 7 |
| MAHA | ViT | 9 | 9 | 9 | 9 | 9 | 9 | 4 | 7 | 9 | 4 | 1 | 5 | | 1 | 9 | 4 | 1 | 7 | | 1 | 9 | 1 | 1 | 1 | 2 |

Table 12: **Comparing Rankings of** $\text{AURC} \to \alpha$ **versus** $\text{AUROC}_f \to \beta$. The fact that these two metrics result in substantially different rankings of CSFs demonstrates the important effect of the CSFs on classifier performance and thus the relevance of the pitfall visualized in Figure 4. In Section 2 we argue that the effects of CSFs on the classifier need to be taken into account for fair comparison (R1), i.e. the ability to prevent silent failures should be assessed by AURC instead of $\text{AUROC}_f$. The color heatmap is normalized per column and classifier (separately for CNN and ViT), while whiter colors depict better scores. "cor" is the average over 5 intensity levels of image corruption shifts. Scores are averaged over 5 runs per experiment on all data sets except 10 runs on CAMELYON-17-Wilds (as recommended by the authors due to high volatility in results) and 2 runs on BREEDS. Abbreviations: ncs: new-class shift (s for semantic, ns for non-semantic), iid: independent and identically distributed, sub: sub-class shift, cor: image corruptions, c10/100: CIFAR-10/100, ti: TinyImagenet.

| | study | iWildCam iid (α β) | iWildCam sub (α β) | iWildCam s-ncs (α β) | BREEDS iid (α β) | BREEDS sub (α β) | CAMELYON iid (α β) | CAMELYON sub (α β) | CIFAR-100 iid (α β) | CIFAR-100 sub (α β) | CIFAR-100 cor (α β) | CIFAR-100 s-ncs c10 (α β) | CIFAR-100 ns-ncs svhn (α β) | CIFAR-100 ns-ncs ti (α β) | CIFAR-10 iid (α β) | CIFAR-10 cor (α β) | CIFAR-10 s-ncs c100 (α β) | CIFAR-10 ns-ncs svhn (α β) | CIFAR-10 ns-ncs ti (α β) | SVHN iid (α β) | SVHN s-ncs (α β) | SVHN ns-ncs c10 (α β) | SVHN ns-ncs c100 (α β) | SVHN ns-ncs ti (α β) |
|---|---|---|---|---|---|---|---|---|---|---|---|---|---|---|---|---|---|---|---|---|---|---|---|---|
| MSR | CNN | 6 6 | 4 5 | 7 7 | 5 5 | 6 6 | 9 9 | 2 2 | 3 5 | 7 10 | 7 8 | 3 2 | 3 4 | 10 10 | 9 5 | 10 6 | 7 9 | 7 8 | 8 9 | 5 5 | 4 11 | 10 10 | 10 10 | 11 11 |
| MLS | CNN | 9 8 | 9 9 | 11 10 | 12 12 | 12 11 | 9 9 | 2 2 | 10 8 | 9 7 | 11 5 | 8 1 | 6 5 | 1 1 | 12 11 | 11 7 | 1 1 | 2 3 | 1 1 | 10 10 | 3 6 | 2 2 | 2 3 | 4 4 |
| PE | CNN | 7 7 | 5 6 | 6 5 | 7 6 | 5 4 | 9 9 | 2 2 | 8 4 | 10 8 | 9 4 | 9 8 | 11 11 | 3 3 | 8 4 | 9 3 | 6 6 | 4 4 | 5 5 | 5 4 | 4 9 | 7 7 | 7 7 | 9 9 |
| MCD-MSR | CNN | 1 1 | 1 1 | 3 6 | 2 2 | 2 4 | 4 4 | 5 6 | 2 3 | 2 3 | 2 2 | 1 3 | 8 10 | 9 9 | 2 3 | 3 4 | 7 7 | 10 10 | 10 10 | 2 2 | 11 5 | 6 6 | 6 6 | 7 7 |
| MCD-MLS | CNN | 8 10 | 8 8 | 4 9 | 11 11 | 11 10 | 7 7 | 8 8 | 9 11 | 5 6 | 5 9 | 7 6 | 2 2 | 5 5 | 11 12 | 5 10 | 1 2 | 4 7 | 2 2 | 9 9 | 8 1 | 1 1 | 1 1 | 1 1 |
| MCD-PE | CNN | 2 3 | 2 3 | 1 3 | 3 3 | 2 2 | 4 4 | 5 6 | 4 6 | 2 4 | 3 6 | 3 4 | 5 6 | 7 7 | 6 7 | 2 5 | 4 4 | 9 9 | 9 8 | 4 6 | 10 3 | 5 5 | 5 5 | 2 2 |
| MCD-EE | CNN | 4 5 | 3 4 | 1 2 | 4 4 | 2 3 | 6 6 | 5 5 | 5 7 | 4 5 | 4 7 | 5 9 | 6 8 | 8 8 | 7 8 | 1 2 | 3 3 | 7 6 | 5 6 | 5 7 | 9 2 | 4 4 | 4 4 | 3 3 |
| MCD-MI | CNN | 2 4 | 6 7 | 8 12 | 5 8 | 7 9 | 8 8 | 9 9 | 7 10 | 6 9 | 6 10 | 5 10 | 4 3 | 6 6 | 10 10 | 6 11 | 10 10 | 12 12 | 12 12 | 8 8 | 12 7 | 9 9 | 9 9 | 6 6 |
| ConfidNet | CNN | 12 12 | 12 12 | 5 4 | 8 7 | 7 7 | 3 1 | 1 1 | 6 1 | 8 1 | 8 1 | 10 6 | 10 9 | 2 2 | 4 1 | 7 1 | 4 5 | 4 5 | 7 7 | 3 3 | 4 10 | 11 11 | 11 11 | 12 12 |
| DG-MCD-MSR | CNN | 5 2 | 7 2 | 9 1 | 1 1 | 1 1 | 2 3 | 11 12 | 1 2 | 1 2 | 1 3 | 1 5 | 9 7 | 11 11 | 3 2 | 4 8 | 11 11 | 11 11 | 11 11 | 1 1 | 1 4 | 8 8 | 8 8 | 8 8 |
| DG-Res | CNN | 11 11 | 10 11 | 12 8 | 10 9 | 10 8 | 1 2 | 10 10 | 11 12 | 12 12 | 10 12 | 11 11 | 1 1 | 4 4 | 5 9 | 8 9 | 12 12 | 3 2 | 3 3 | 11 11 | 2 8 | 3 3 | 3 2 | 5 5 |
| Devries et al. | CNN | 10 9 | 11 10 | 10 11 | 9 10 | 9 12 | 12 12 | 12 11 | 12 9 | 11 11 | 12 11 | 12 12 | 12 12 | 12 12 | 1 6 | 12 12 | 7 8 | 1 1 | 3 4 | 12 12 | 7 12 | 12 12 | 12 12 | 10 10 |
| | | | | | | | | | | | | | | | | | | | | | | | | |
| MSR | ViT | 2 2 | 2 2 | 7 3 | 5 5 | 1 6 | 5 5 | 2 2 | 5 6 | 1 5 | 2 2 | 6 7 | 3 3 | 9 9 | 5 6 | 2 2 | 3 5 | 3 4 | 8 9 | 5 6 | 8 9 | 9 8 | 9 8 | 9 9 |
| MLS | ViT | 3 7 | 3 8 | 8 2 | 8 8 | 7 8 | 8 8 | 1 1 | 8 8 | 7 8 | 3 7 | 2 1 | 1 1 | 7 7 | 8 8 | 3 5 | 3 3 | 2 2 | 7 4 | 8 8 | 7 7 | 3 3 | 4 3 | 3 2 |
| PE | ViT | 1 1 | 1 1 | 6 1 | 5 5 | 1 5 | 5 5 | 2 2 | 5 5 | 1 6 | 1 1 | 4 5 | 2 2 | 8 8 | 5 6 | 1 1 | 3 4 | 3 3 | 8 8 | 5 5 | 8 8 | 7 6 | 8 6 | 8 8 |
| MCD-MSR | ViT | 6 5 | 6 6 | 4 7 | 1 1 | 3 2 | 1 1 | 1 1 | 5 3 | 8 8 | 8 8 | 8 8 | 1 1 | 1 1 | 8 7 | 7 8 | 4 6 | 3 3 | 5 5 | 3 5 | 6 7 | 6 7 | 6 7 | 6 6 |
| MCD-MLS | ViT | 8 8 | 8 7 | 5 8 | 7 7 | 8 7 | 7 7 | 9 9 | 9 7 | 8 7 | 9 8 | 3 3 | 4 4 | 2 1 | 5 4 | 2 2 | 1 1 | 2 2 | 7 7 | 7 7 | 3 2 | 2 2 | 2 2 | 1 1 |
| MCD-PE | ViT | 4 3 | 4 3 | 1 4 | 2 2 | 3 1 | 1 1 | 5 5 | 2 2 | 3 1 | 6 4 | 6 6 | 7 7 | 4 4 | 1 2 | 7 5 | 7 7 | 6 6 | 4 4 | 1 1 | 3 4 | 5 5 | 5 5 | 5 5 |
| MCD-EE | ViT | 6 6 | 5 4 | 1 5 | 2 2 | 3 4 | 1 1 | 5 7 | 2 3 | 6 4 | 5 3 | 4 4 | 3 3 | 1 2 | 6 3 | 6 6 | 5 5 | 3 3 | 1 1 | 2 3 | 4 4 | 3 4 | 4 4 | 4 4 |
| MCD-MI | ViT | 4 4 | 6 5 | 1 6 | 2 4 | 3 3 | 4 4 | 8 8 | 4 4 | 3 2 | 8 6 | 9 9 | 6 6 | 4 4 | 9 9 | 9 9 | 6 7 | 4 4 | 3 6 | 4 4 | 7 9 | 7 9 | 7 9 | 7 7 |
| MAHA | ViT | 9 9 | 9 9 | 9 9 | 9 9 | 9 9 | 4 4 | 7 9 | 9 9 | 4 9 | 1 2 | 5 6 | 1 2 | 5 6 | 9 9 | 4 9 | 1 1 | 7 7 | 1 1 | 9 9 | 1 1 | 1 1 | 1 1 | 2 3 |

Table 13: **Comparing Rankings of $\mathrm{AUROC}_f$ based on the current OoD-protocol ("original" → O) versus the proposed modification of dismissing inlier misclassifications ("proposed" → P).** The fact that these two protocols lead to considerably different rankings of CSFs in many scenarios demonstrates the importance of the effect of inlier misclassifications on OoD-D evaluation and thus the importance of the proposed modification (visualized in Figure 5). The color heatmap is normalized per column and classifier (separately for CNN and ViT), while whiter colors depict better scores. Scores are averaged over 5 runs per experiment on all data sets. Abbreviations: ncs: new-class shift (s for semantic, ns for non-semantic), c10/100: CIFAR-10/100, ti: TinyImagenet.

| | | iWildCam | | CIFAR-100 | | | | | | CIFAR-10 | | | | | | SVHN | | | | | | |
| | study | s-ncs | | s-ncs | | ns-ncs | | | | s-ncs | | ns-ncs | | | | s-ncs | | ns-ncs | | | | |
| | ncs-data set | | | c10 | | svhn | | ti | | c100 | | svhn | | ti | | | | c10 | | c100 | | ti |
| | ood protocol | P | O | P | O | P | O | P | O | P | O | P | O | P | O | P | O | P | O | P | O | P | O |
|---|---|---|---|---|---|---|---|---|---|---|---|---|---|---|---|---|---|---|---|---|---|---|---|
| MSR | CNN | 7 | 8 | 2 | 7 | 4 | 5 | 10 | 10 | 9 | 10 | 8 | 8 | 9 | 9 | 11 | 11 | 10 | 11 | 10 | 10 | 11 | 11 |
| MLS | CNN | 10 | 9 | 1 | 3 | 5 | 6 | 1 | 1 | 1 | 2 | 3 | 3 | 1 | 1 | 6 | 6 | 2 | 2 | 3 | 3 | 4 | 3 |
| PE | CNN | 5 | 7 | 8 | 10 | 11 | 11 | 3 | 2 | 6 | 6 | 4 | 4 | 5 | 6 | 9 | 10 | 7 | 6 | 7 | 6 | 9 | 10 |
| MCD-MSR | CNN | 6 | 10 | 3 | 6 | 10 | 9 | 9 | 9 | 7 | 7 | 10 | 10 | 10 | 10 | 5 | 5 | 6 | 7 | 6 | 7 | 7 | 8 |
| MCD-MLS | CNN | 9 | 4 | 6 | 1 | 2 | 3 | 5 | 6 | 2 | 1 | 7 | 6 | 2 | 2 | 1 | 1 | 1 | 1 | 1 | 1 | 1 | 1 |
| MCD-PE | CNN | 3 | 3 | 4 | 2 | 6 | 4 | 7 | 7 | 4 | 4 | 9 | 9 | 8 | 7 | 3 | 3 | 5 | 5 | 5 | 5 | 2 | 2 |
| MCD-EE | CNN | 2 | 2 | 9 | 4 | 8 | 7 | 8 | 8 | 3 | 3 | 6 | 5 | 6 | 5 | 2 | 2 | 4 | 4 | 4 | 4 | 3 | 4 |
| MCD-MI | CNN | 12 | 12 | 10 | 5 | 3 | 2 | 6 | 5 | 10 | 8 | 12 | 12 | 12 | 12 | 7 | 7 | 9 | 9 | 9 | 9 | 6 | 6 |
| ConfidNet | CNN | 4 | 5 | 6 | 11 | 9 | 10 | 2 | 3 | 5 | 5 | 5 | 7 | 7 | 8 | 10 | 9 | 11 | 12 | 11 | 12 | 12 | 12 |
| DG-MCD-MSR | CNN | 1 | 5 | 5 | 8 | 7 | 8 | 11 | 11 | 11 | 12 | 11 | 11 | 11 | 11 | 4 | 4 | 8 | 10 | 8 | 8 | 8 | 9 |
| DG-Res | CNN | 8 | 1 | 11 | 9 | 1 | 1 | 4 | 4 | 12 | 11 | 2 | 2 | 3 | 3 | 8 | 8 | 3 | 3 | 2 | 2 | 5 | 5 |
| Devries et al. | CNN | 11 | 11 | 12 | 12 | 12 | 12 | 12 | 12 | 8 | 9 | 1 | 1 | 4 | 4 | 12 | 12 | 12 | 8 | 12 | 11 | 10 | 7 |
| MSR | ViT | 3 | 8 | 7 | 7 | 3 | 4 | 9 | 9 | 5 | 5 | 4 | 4 | 9 | 8 | 9 | 9 | 8 | 7 | 8 | 7 | 9 | 9 |
| MLS | ViT | 2 | 4 | 1 | 2 | 1 | 1 | 7 | 7 | 3 | 3 | 2 | 2 | 4 | 3 | 7 | 7 | 3 | 3 | 3 | 3 | 2 | 2 |
| PE | ViT | 1 | 7 | 5 | 5 | 2 | 3 | 8 | 8 | 4 | 4 | 3 | 3 | 8 | 7 | 8 | 8 | 6 | 6 | 6 | 6 | 8 | 6 |
| MCD-MSR | ViT | 7 | 5 | 8 | 8 | 8 | 8 | 5 | 5 | 8 | 8 | 8 | 8 | 6 | 6 | 5 | 5 | 7 | 8 | 7 | 7 | 6 | 6 |
| MCD-MLS | ViT | 8 | 1 | 3 | 3 | 4 | 2 | 1 | 2 | 2 | 2 | 1 | 1 | 2 | 2 | 2 | 2 | 2 | 2 | 2 | 2 | 1 | 2 |
| MCD-PE | ViT | 4 | 3 | 6 | 6 | 7 | 7 | 4 | 4 | 7 | 7 | 6 | 7 | 4 | 5 | 4 | 4 | 5 | 5 | 5 | 5 | 5 | 5 |
| MCD-EE | ViT | 5 | 2 | 4 | 4 | 5 | 6 | 3 | 3 | 6 | 6 | 5 | 5 | 3 | 4 | 3 | 3 | 4 | 4 | 4 | 4 | 4 | 4 |
| MCD-MI | ViT | 6 | 6 | 9 | 9 | 9 | 9 | 6 | 6 | 9 | 9 | 9 | 9 | 7 | 9 | 6 | 6 | 9 | 9 | 9 | 9 | 7 | 8 |
| MAHA | ViT | 9 | 9 | 2 | 1 | 6 | 5 | 2 | 1 | 1 | 1 | 7 | 6 | 1 | 1 | 1 | 1 | 1 | 1 | 1 | 1 | 3 | 1 |

# H   QUALITATIVE ANALYSIS OF FAILURE CASES

We categorize two types of errors for confidence scores: Overconfidence, i.e. assigning high confidence to wrong predictions, and underconfidence, i.e. assigning low confidence to correct predictions. Figure 9a) shows the three most overconfident and underconfident test cases of the MSR confidence score on CIFAR-100 per distribution shift (since test cases subject to new-class shift are incorrect by default, there are no underconfident predictions). Looking at overconfident predictions, on the i.i.d. test set we see some expected confusions between semantically close categories. Images under corruption shift are subjectively hard to classify even for the human eye. The sub-class shift reveals an ambiguity in the labels rather than a failure of the confidence score containing objects of two valid categories, fish and people, indicating that superclasses in CIFAR-100 have not been specifically designed for sub-class shift by e.g. excluding such cases (Figure 9b looks at failure cases of BREEDS-Entitiy-13 which has been curated for meaningful sub-class shift). On the new-class shifts, we observe the expected behaviour of predicting the training categories most semantically similar to the unknown categories. Looking at underconfident cases reveals some interesting shortcut learning (Geirhos et al., 2020): The fact that the two racoon images received low confidence despite showing their faces indicates that the model relies on spurious features perhaps related to the background. Similarly, the fox image on the corruption tests may have received low confidence due to the artificial frame. Further, the colored pixel noise in the hamster image as a source of confusion hints at the mode relying on local texture patterns rather than global semantic features.

While the previous qualitative study only showcases confidence based on MSR, the proposed unified protocol facilitates to study arbitrary confidence scoring functions in detail and across different distribution shifts. Thus, in Figure 1b) we study failure cases of the Predictive Entropy obtained from Monte Carlo Dropout (MCD-PE) on the BREEDS-Entity-13 data set. Entropy scores are reversed and

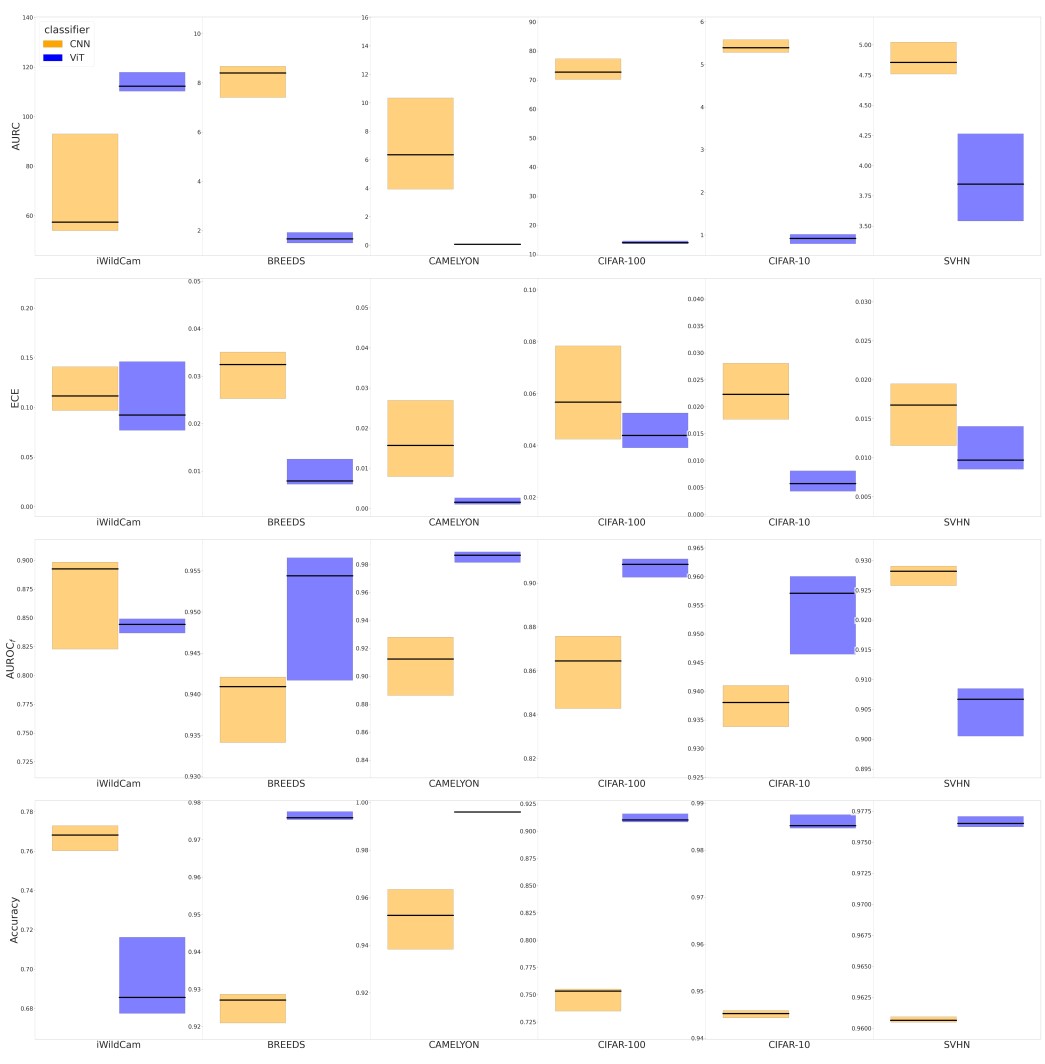

Figure 8: **Comparison of CNN versus ViT-based Classifier.** Boxplots show median (black line) and quartiles, and include all CSFs and studies shown in Table 1

scaled to $[0, 1]$ for interpretability. Note, how an "external" confidence score like entropy can be zero even on the predicted class (such decoupling would be inherently impossible with MSR). Looking at overconfident cases again reveals label ambiguities on the i.i.d. test set rather than actual failures of confidence scoring. Examples are a bird in front of a monitor ("monitor" is a training sub-class to "equipment"), a woman wearing a scarf on top of a poncho ("poncho" is a training sub-class to "garment"), or a vehicle behind a fence ("fence" is a sub-class to man-made structure). Further, this study reveals interesting shortcut learning such as focusing on knitting patterns ("garment" vs. "accessory"), or a dog being predicted as unlikely to appear in front of the water. A really hard categorization task is posed by the image showing hot air balloons, as they are confused with volleyballs (training sub-class to "equipment").

# I    TECHNICAL FORMULATION OF CLASSIFIER-OUTPUT BASED CONFIDENCE SCORING FUNCTIONS

The majority of confidence scoring functions explored in this study is based on one of the well-established methods for quantifying the uncertainty of a classifier's prediction. While strictly

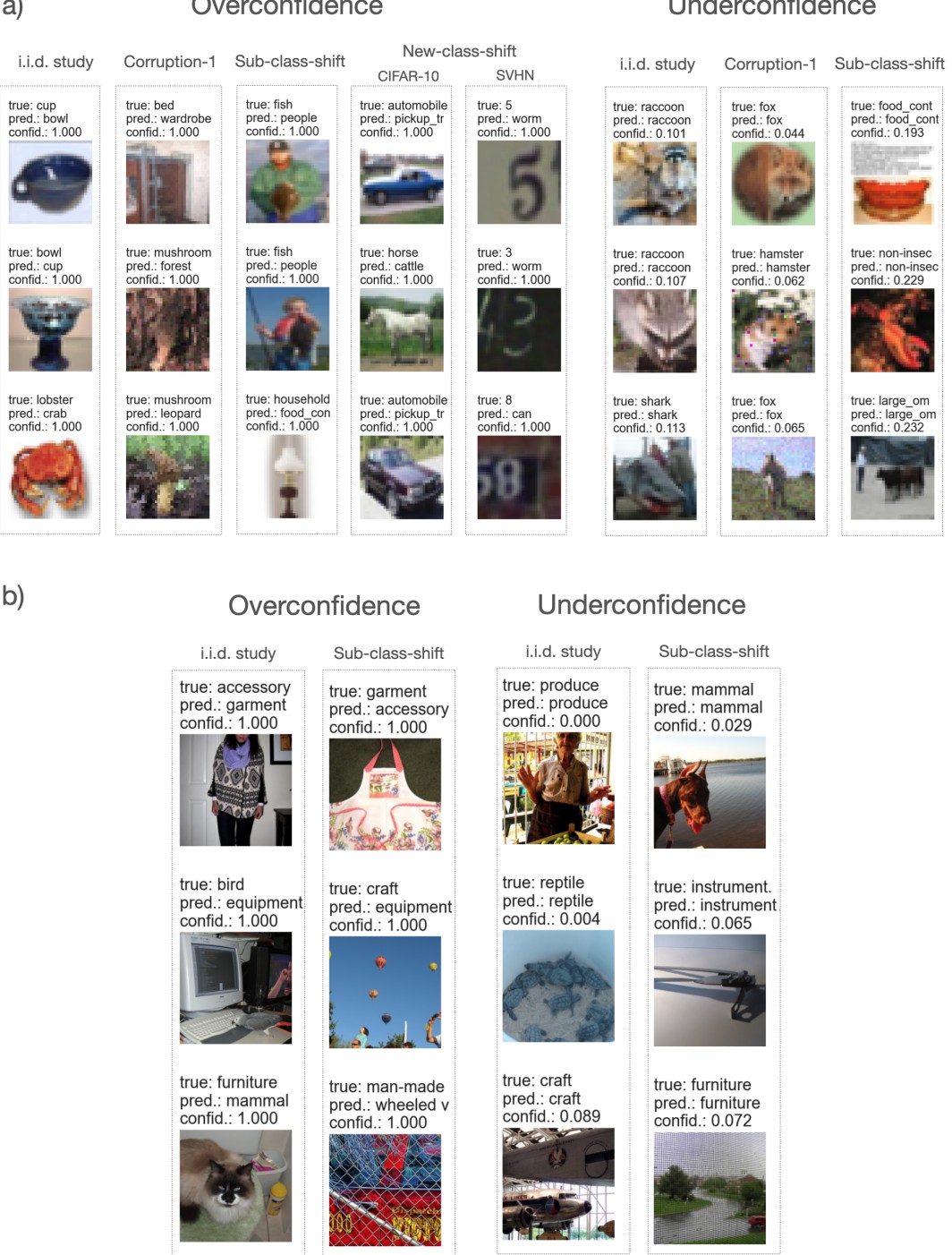

Figure 9: Qualitative study of failure cases in failure detection. **a)** The three most overconfident and underconfident test cases of MSR for different distribution shifts on CIFAR-100. b) The three most overconfident and underconfident test cases of Predictive Entropy obtained by Monte Carlo Dropout (MCD-PE) on BREEDS-Entitiy-13.

proper scoring rules such as NLL or Brier Score only assess performance related to failure detection for one of those measures, MSR, the protocol proposed in this study enables comparison across arbitrary measures. While PE is often associated with capturing a prediction's total uncertainty ((Smith and Gal, 2018; Depeweg et al., 2018; Malinin and Gales, 2018)), MCD-EE is said to capture the uncertainty inherent in the data (or *aleatoric uncertainty*). Consequentially, subtracting the latter from the former, i.e. the mutual information (MCD-MI) between the parameters $\boldsymbol{\theta}$ and the categorical label $y$, is often associated with only capturing the model uncertainty (a form of *epistemic uncertainty*). The Maximum Logit Score (MLS), proposed by Vaze et al. (2022), uses the magnitude of the logits which they argue carries information relevant for OoD-D, where greater logits should correspont to more certain predictions.

**Maximum Softmax Response (MSR):**

$$\mathcal{P} = \max_c P(y = c | \boldsymbol{x}^*; \mathcal{D}), \tag{34}$$

where $c$ represents the different classes.

**Predictive Entropy (PE):**

$$\mathcal{H}[\mathrm{P}(y|\boldsymbol{x}^*; \mathcal{D})] = -\sum_{c=1}^{C} P(y = c | \boldsymbol{x}^*; \mathcal{D}) \cdot \ln(P(y = c | \boldsymbol{x}^*; \mathcal{D}) \tag{35}$$

**Maximum Logit Score (MLS):**

$$\max_c f(\boldsymbol{x}^*, \mathcal{D}), \tag{36}$$

where $f$ is the function producing the logits that are used to create the probability vector $P(y|\boldsymbol{x}^*, \mathcal{D}) = \mathrm{softmax}(f(\boldsymbol{x}^*, \mathcal{D}))$.

**Maximum Softmax Response obtained from Monte Carlo Dropout (MCD-MSR):**

$$\mathcal{P} = \max_c \mathbb{E}_{\mathrm{p}(\boldsymbol{\theta}|\mathcal{D})}[\mathrm{P}(\omega_c | \boldsymbol{x}^*; \boldsymbol{\theta})], \tag{37}$$

where $\boldsymbol{\theta}$ are the model parameters.

**Predictive Entropy obtained from Monte Carlo Dropout (MCD-PE):**

$$\mathcal{H}[\mathbb{E}_{\mathrm{p}(\boldsymbol{\theta}|\mathcal{D})}[\mathrm{P}(y|\boldsymbol{x}^*; \boldsymbol{\theta})]], \tag{38}$$

**Expected Entropy obtained from Monte Carlo Dropout (MCD-EE):**

$$\mathbb{E}_{\mathrm{p}(\boldsymbol{\theta}|\mathcal{D})}[\mathcal{H}[\mathrm{P}(y|\boldsymbol{x}^*; \boldsymbol{\theta})]] \tag{39}$$

**Mutual Information obtained from Monte Carlo Dropout (MCD-MI):**

$$I(y; \boldsymbol{\theta}|) = \mathcal{H}[\mathbb{E}_{\mathrm{p}(\boldsymbol{\theta}|\mathcal{D})}[\mathrm{P}(y|\boldsymbol{x}^*; \boldsymbol{\theta})]] - \mathbb{E}_{\mathrm{p}(\boldsymbol{\theta}|\mathcal{D})}[\mathcal{H}[\mathrm{P}(y|\boldsymbol{x}^*; \boldsymbol{\theta})]] \tag{40}$$

**Maximum Logit Score obtained from Monte Carlo Dropout (MCD-MLS):**

$$\max_c \mathbb{E}_{\mathrm{p}(\boldsymbol{\theta}|\mathcal{D})}[f(\boldsymbol{x}^*, \boldsymbol{\theta})], \tag{41}$$

where $f$ is the function producing the logits that are used to create the probability vector $P(y|\boldsymbol{x}^*, \boldsymbol{\theta}) = \mathrm{softmax}(f(\boldsymbol{x}^*, \boldsymbol{\theta}))$.

## J    REPRODUCIBILITY AND DEVIATIONS FROM ORIGINAL BASELINE CONFIGURATIONS

As described in Section E.3, a meaningful evaluation of confidence scoring methods for failure detection requires deviating from original configurations of baselines in order to assimilate the underlying classifier across compared methods. Since re-implementing and re-configuring baseline methods is prone to inducing biases in comparative studies and respective conclusions (Lipton and Steinhardt, 2018), in this section, we want to explain all deviations in detail and provide evidence that baselines perform on a par with or superior to the original versions under our configuration.

In Table 14, we compare our configuration against the original configuration using the results reported on CIFAR-10. We deviate from the original configuration in using additional data augmentation in the form of a cutout, decaying the learning rate via cosine scheduler, training a VGG-13 model instead of VGG-16, and not using dropout for training (as selected by our model selection). The hyperparameter $o$ has been selected as 2.2 on this data set, which is identical to the original configuration. The comparison shows, that our results are substantially better than the original configuration, mainly due to a better classifier accuracy.

Table 14: Comparing our DeepGamblers configuration to the original configuration (as proposed in Liu et al. (2019)) on CIFAR-10. The table shows risk scores at predefined coverages. Lower is better.

| Coverage | our configuration | original configuration |
|----------|-------------------|------------------------|
| 1.00 | **4.94** | 6.12 |
| 0.95 | **2.69** | 3.49 |
| 0.90 | **1.37** | 2.19 |
| 0.85 | **0.82** | 1.09 |
| 0.80 | **0.54** | 0.66 |
| 0.75 | **0.43** | 0.52 |

## J.2  DEVRIES ET AL.

The only substantial deviation from the original protocol is the replacement of the proposed MultiStep learning rate scheduler with our cosine scheduler (the latter is more aggressive with overall higher learning rates). As Figure 10 shows, this deviation has a negative effect on the OoD-detection performance only when training on CIFAR-10 and testing on TinyImagenet. There, our configuration only achieves 0.965 AUROC instead of 97.0 AUROC from the original paper. In all other settings, the cosine scheduler results in superior performance. As part of the assimilation of the training configuration across compared methods, we increased the number of training epochs for Devries et al. from 200 to 250. Since performance did not benefit from this increase in training time, we believe there is no fairness issue with DeepGamblers and ConfidNet requiring additional training stages leading to overall more training exposure (see Table 4). We did not include additional confidence measures such as MSR, PE, or any MCD variations based on the Devries et al. learning objective, as we found no interesting synergies to occur (this is in contrast to DeepGamblers and ConfidNet).

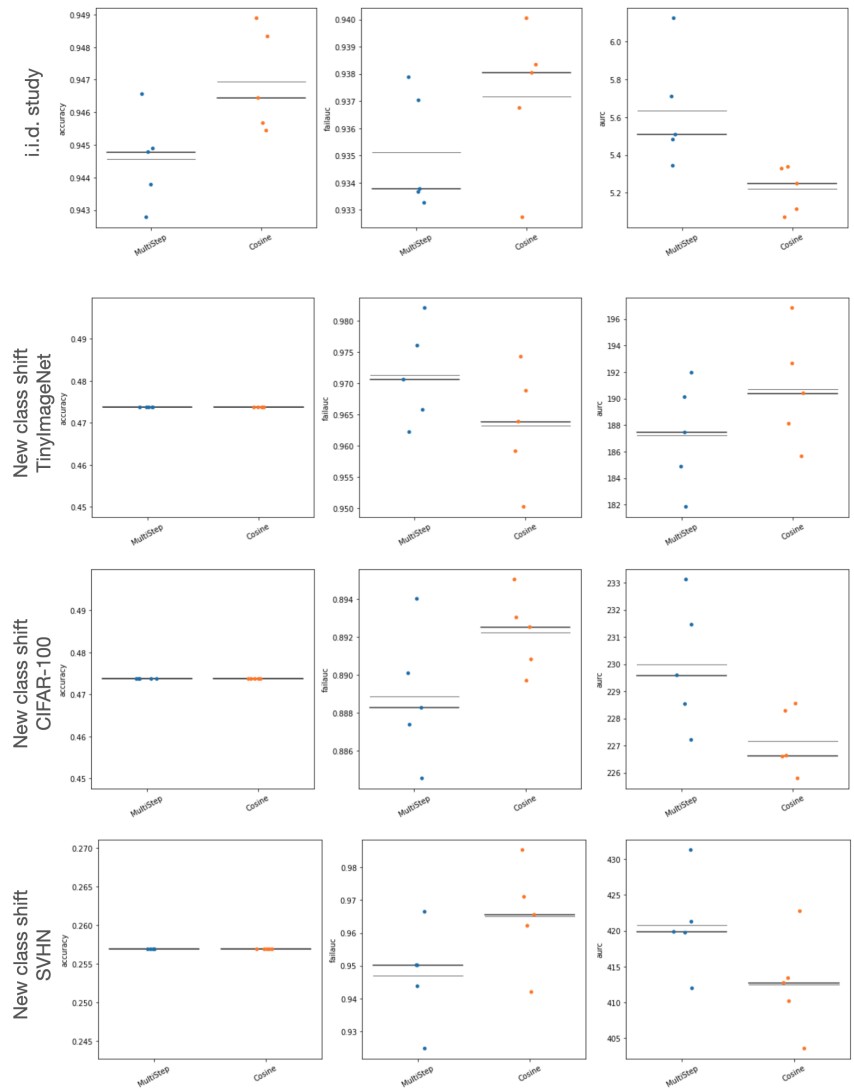

Figure 10: Ablation study comparing the originally proposed Multistep scheduler against the Cosine scheduler for the confidence scores from Devries et al. trained on CIFAR-10. One dot denotes one run, the thin line denotes the mean across runs and the thick line denotes the median across runs.

## J.3 CONFIDNET

The original authors propose a three-stage epoch selection, first selecting the best classifier according to validation accuracy, second selecting the best ConfidNet training epoch according to fail-average-precision (with failure defined as the positive label), and third selecting the best ConfidNet finetuning epoch according to fail-average-precision (Corbière et al., 2019). We found results to be very volatile across single epochs, especially for fail-average precision. Since reducing the validation set to 1000 cases in the process of assimilating training protocols (the original configuration used 5000 cases from the training data) amplified volatility further, we ran an ablation study comparing the proposed model selection to simply taking the last epoch at each stage. Figure 11 shows the results, indicating that (at least for our smaller validation set), choosing the latest epoch performs superior to epoch selection. We further deviate from the original configuration by using VGG-13 instead of VGG-16 and applying cutout in addition for data augmentation. Comparing our results to the ones in the original publication, we vastly improve classifier accuracy and thus AURC scores on SVHN and CIFAR-10/100, although it is unclear how much of this increase is to be attributed to our configuration and how much to the different training and validation splits.

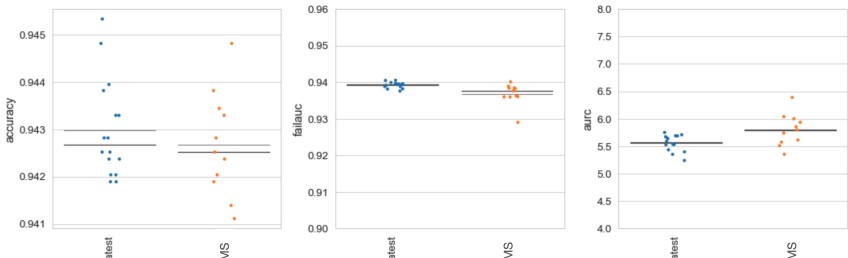

Figure 11: Ablation study for the three-staged model selection proposed in the original ConfidNet configuration. One dot denotes one run and evaluation on the CIFAR-10 i.i.d. test set, the thin line denotes the mean across runs and the thick line denotes the median across runs. MS: Model selection, latest: select the latest epoch at each training stage.

## J.4 MAXIMUM LOGIT SCORE

Since Vaze et al. (2022) do not include the ViT architecture in their evaluation, we only compare CNN results. While Vaze et al. (2022) use the VGG-32 classifier, we use different classifiers depending on the data set as discussed in Appendix E.2. On each data set we use the same classifier for MLS evaluation as for the other methods. The authors employ a variety of improvements to their baseline strategy of which we only use some: we do use cosine annealing to schedule learning rates, but do not employ label smoothing or ensembling to ensure fair comparison with other CSFs. Thus, we expect our results to lie between their baseline and their improved baseline. Table 15 compares our configuration to theirs as measured by $AUROC_f$ on the SVHN dataset in the original open-set setting, and confirms our expected performance relations.

Table 15: Comparing our classifier configuration to the configuration proposed by Vaze et al. (2022) as measured by $AUROC_f$ on the SVHN dataset in the open-set setting. Baseline MLS and Baseline MSR+ use the aforementioned improvements to the training strategy.

| Method | $AUROC_f * 100 \uparrow$ |
|---|---|
| Baseline MSR (Vaze et al., 2022) | 88.6 |
| MSR (ours) | 94.0 |
| MLS (ours) | 94.3 |
| MSR (improved baseline) (Vaze et al., 2022) | 96.0 |
| MLS (improved baseline) (Vaze et al., 2022) | 97.1 |

## J.5 MAHALANOBIS SCORE

Fort et al. (2021) compare different vision transformer architectures, due to computational constraints we only consider the ViT-B/16 architectue. Since the original publication does not report how results have been obtained, i.e. whether multiple runs have been performed and how final results were selected from these runs, we report both our best run as well as an average over five runs, selected by best accuracy. Table 16 compares our implementation to theirs as measured by $AUROC_f$ trained on CIFAR-10/CIFAR-100 using CIFAR-100/CIFAR-10 as the out-of-distribution dataset respectively. We see that our average is slightly below the original results, but results are very volatile with high standard deviations over runs. Our best run results are en par with the original results. Equivalently to original results, MAHA score are consistently higher compared to MSR.

Table 16: Comparing our ViT-B/16 implementation to the one by Fort et al. (2021) as measured by $\text{AUROC}_f$ when fine-tuned on CIFAR-10/CIFAR-100 using CIFAR-100/CIFAR-10 as the out-of-distribution dataset respectively.

| ID to OD | MAHA $\text{AUROC}_f * 100 \uparrow$ | MSR $\text{AUROC}_f * 100 \uparrow$ |
|---|---|---|
| CIFAR-10 $\rightarrow$ CIFAR-100 (ours, mean) | $97.99 \pm 0.31$ | $96.71 \pm 0.39$ |
| CIFAR-10 $\rightarrow$ CIFAR-100 (ours, best) | 98.49 | 97.69 |
| CIFAR-10 $\rightarrow$ CIFAR-100 (Fort et al., 2021) | 98.42 | 97.68 |
| CIFAR-100 $\rightarrow$ CIFAR-10 (ours, mean) | $91.13 \pm 7.86$ | $89.02 \pm 4.96$ |
| CIFAR-100 $\rightarrow$ CIFAR-10 (ours, best) | 95.23 | 92.50 |
| CIFAR-100 $\rightarrow$ CIFAR-10 (Fort et al., 2021) | 95.53 | 91.89 |

