# OpenReview forum: "A Call to Reflect on Evaluation Practices for Failure Detection in Image Classification"
_ICLR.cc/2023/Conference — ICLR 2023 notable top 5%_

### Official Review · Reviewer_7dx4 · 2022-10-24

**Confidence:** 3
**Correctness:** 4
**Technical Novelty And Significance:** 3
**Empirical Novelty And Significance:** 4
**Recommendation:** 8

**Clarity, Quality, Novelty And Reproducibility:**

This paper is of high clarity and quality. Although it does not propose a new approach/technique/method, I still believe it has considerable novelty.

**Strength And Weaknesses:**

Strengths
1. I strongly encourage the intention stated by the authors: to articulate a call to various interrelated yet isolated communities to acknowledge the shortcomings of current practices and adapt to a more reasonable practice that not only better fit their alleged purpose but also help bring the isolated fields together.
2. Section 2 is a deliberate and eye-catching section. The authors discussed three pitfalls of the existing methods and derived the corresponding point-by-point requirements for a desired evaluation protocol for failure detection. Among them, I particularly like (1) the second pitfall where the authors drew out attention towards different sources of classification failure and pointed out that existing methods often cover a subset of them (also illustrated in Figure 1). (2) the third pitfall where most relevant research claim failure detection as their purpose but in fact are evaluated on outlier detection, causing problems due to the mismatch (also illustrated in Figure 2).
3. I also like that the authors pointed out an easily neglected issue: rounding errors during softmax operation may largely affect the resulting evaluation metrics.

Weaknesses
1. Personally I believe reorganizing section 4.2 would be beneficial. For example, consider organizing the CSFs to be compared by their subfields (MisD, SC, PUQ, OoD-D) if possible?
2. Table 1 is a bit difficult for me to quickly process — I understand what is displayed but my brain is not well trained to instantly make sense of this representation. Is this just me or is it a common inconvenience? If it’s that latter, I would suggest adding a column for each dataset showing the “sum of rank order for the current method across all evaluations in this dataset”.

Minor Things
1. Typo? Section 2, Pitfall 2, last sentence. “…it is not realistic to exclusively assume classification failures from label-altering shifts **an** no failures caused by label-preserving shifts.”
2. Typo? Section 4.3, Under heading “Different types of uncertainty are empirically not distinguishable.“ 3rd line. “…we are interested in the **extend** to which such relations can be confirmed…”
3. Is there any reason $y_{f,i}$ in Eq. 24 is highlighted?

**Summary Of The Paper:**

With the advancements of machine learning and boosting availability of compatible hardware as well as cloud services, machine learning-based classification systems have been deployed in numerous scenarios. Regardless of the specific application — some more critical than others — it is essential to detect the failure cases in a timely manner in order to quickly iterate on targeted improvements.

The authors pointed out that currently the dominant means for such failure detection is through checking some type of confidence scores, either directly produced by the model as an output, or indirectly calculated through some other heuristics (e.g., recall the notion of uncertainty in Focal Loss). The authors claimed that these different approaches are aiming to solve the same failure detection problem yet through distinctive methods which led to diverging protocols of various subfields (misclassification detection, out-of-distribution detection, selective classification, predictive uncertainty quantification). The authors then systematically analyzed the existing methods to reveal their pitfalls, and proposed a holistic and realistic approach for failure detection.

The proposed metric is the Area under the Risk-Coverage Curve (AURC), which is proposed in Geifman et al. This metric assess the rate of silent failures of a classifier across different failure filtering thresholds. The authors claimed that the metric avoids all pitfalls they revealed, and recommended wide adoption across all subfields of research within classification failure detection. The authors further delineated the modification of protocols needed to adapt to the new evaluation approach from these aforementioned subfields.

**Summary Of The Review:**

This paper encouraged researchers to view multiple subfields (misclassification detection, out-of-distribution detection, selective classification, predictive uncertainty quantification) under the umbrella of classification failure detection through a unified perspective, and proposed improvements to the problem statements and walked through the design of a generic evaluation protocol. More remarkably, the authors showed that a simple baseline can match or beat the state-of-the-arts in these subfields on the unified benchmark, since these SOTA techniques only perform in their selective subfield and miss the generalizability in the bigger picture. I like the scope, vision and insight from the authors, and would recommend for acceptance of this paper.

---

> ### Author Response · Authors · 2022-11-15
> **Reply to Reviewer 7dx4**
>
> Thank you for your time and feedback. Below we answer your questions and concerns.
>
> --------
>
> R: Personally I believe reorganizing section 4.2 would be beneficial. For example, consider organizing the CSFs to be compared by their subfields (MisD, SC, PUQ, OoD-D) if possible?
>
> A: Thank you for pointing this out. We structured the compared methods as suggested.
>
> --------
>
> R: Table 1 is a bit difficult for me to quickly process — I understand what is displayed but my brain is not well trained to instantly make sense of this representation. Is this just me or is it a common inconvenience? If it’s that latter, I would suggest adding a column for each dataset showing the “sum of rank order for the current method across all evaluations in this dataset”.
>
> A: Thank you for this feedback. Finding a good visualization of the results of this large-scale study has been quite a journey. We tried several ranking schemes, but the problem with ranking results is that important information is discarded, specifically, the notion to distinguish between “similar performance” and “substantially different performance” of two methods, which we consider quite important for method assessment. When aggregating these rankings over different distribution shift scenarios (resulting in one ranking per data set), as suggested, rankings further become dependent on how many shifts are considered per data set: For instance, the weighting of the i.i.d. performance in the final ranking depends on the number of considered new-class shifts, which varies substantially across data sets and makes the resulting ranking very hard to interpret. We are afraid that there exists no satisfying shortcut to parsing the provided information and still consider the current table to be the best option for visualizing results since no information is lost while at the same time the color heat map can serve as a visual facilitator for method ranking. However, we agree that it takes a few moments to parse the information. To follow your suggestion and help readers get a quick overview of the results, we now added a second version of this table (Table 11) featuring CSF rankings which is linked directly from the main table caption. We comment on the downsides of ranking information in the caption of this new table.
>
> --------
>
> R: Typo Section 2, Pitfall 2, last sentence. “…it is not realistic to exclusively assume classification failures from label-altering shifts an no failures caused by label-preserving shifts.” Typo Section 4.3, Under heading “Different types of uncertainty are empirically not distinguishable.“ 3rd line. “…we are interested in the extend to which such relations can be confirmed…” Is there any reason  yf,i in Eq. 24 is highlighted?
>
> A: Thank you for pointing out these mistakes. We corrected the Type-O and removed the highlighting as it led to confusion.

---

> > ### Comment · Reviewer_7dx4 · 2022-12-05
> > **Response to authors' comments**
> >
> > Thank you. I don't have any more concerns at the moment.

---

### Official Review · Reviewer_JKia · 2022-10-24

**Confidence:** 2
**Correctness:** 4
**Technical Novelty And Significance:** 4
**Empirical Novelty And Significance:** 4
**Recommendation:** 10

**Clarity, Quality, Novelty And Reproducibility:**

I believe the work is high quality and well informed. I do not know the literature.

The perspective and approach are thoughtful.

The authors indicate full reproducibility via open codebases.

**Strength And Weaknesses:**

(strengths)

The paper evidently springs from a substantial understanding of and experience with the topic.

The proposed criteria are thoughtful and offer, agree or disagree, a good point of departure for discussion.

The "step back and look in the mirror" approach is welcome.

The findings section is super interesting.

(weaknesses)

Sometimes the paper covers ground too fast for me (too condensed).

**Summary Of The Paper:**

The paper offers a set of criteria (and a metric area under risk coverage curve) to compare different flavors of failure detection. It gives experiments  that compare different methods according to these criteria.

**Summary Of The Review:**

The proposed criteria are thoughtful and offer, agree or disagree, a good point of departure for discussion.

The "step back and look in the mirror" approach is welcome.

The paper combines many methods that are usually silo'ed.

The findings section is super interesting.

The Conclusions paragraph is wonderful.

---

> ### Author Response · Authors · 2022-11-15
> **Reply to Reviewer JKia**
>
> Thank you for your time and feedback. Below we answer to your concern.
>
> -----
>
> R: Sometimes the paper covers ground too fast for me (too condensed).
>
> A: Thank you for this feedback. Within the possibilities of the space limitations that we have already reached, we added some clarifications to the paper aiming to enhance the reading experience:  We added a visualization for Pitfall 1, thus all three described pitfalls are now visualized (P1 -> Figure 4, P2 -> Figure 1, P3 -> Figure 3) which we hope will help readers to follow the paper. For further clarification, we now provide a simpler ranking version of the main results in Table 11 and a new section G.1 connecting empirical findings to the pitfalls and requirements in Section 2.

---

> > ### Comment · Reviewer_JKia · 2022-11-22
> > **response to authors' comments**
> >
> > Thank you to the other reviewers, who had insights I totally missed.
> >
> > Thank you to the authors for their thoughtful responses.
> >
> > I believe, with poSv, that this paper would be a fine contribution to ICLR.

---

### Official Review · Reviewer_poSv · 2022-10-25

**Confidence:** 3
**Correctness:** 4
**Technical Novelty And Significance:** 1
**Empirical Novelty And Significance:** 4
**Recommendation:** 8

**Clarity, Quality, Novelty And Reproducibility:**

The paper offers a number of valuable insights into failure analysis, confidence scoring and evaluation practices of classifiers. An extensive evaluation  is generated over a number of public benchmarks. I believe the paper would be very valuable to the research community because it brings up overlooked points in evaluation protocols. A detailed code repository is included.

However, the manuscript is relatively difficult to follow and understand.


**Strength And Weaknesses:**

The key strength of the paper is the thorough discussion of evaluation of model classification failures, which is an important and often overlooked part of ML research. The key conclusion is concisely stated in the conclusion: a need for greater introspection and possibly less novelty.

The key weakness is that the paper is difficult to follow. It is not clear how the pitfalls discussed in Section 2 are evaluated or studied in the remainder of the paper. A number of relevant method definitions are found in the appendix, rather than the main paper. In addition, definitions of techniques is often found after the fact. The paper would benefit from greater organization and extraction of key findings and results that support these conclusions. In Table 2, a number of results are reported, however, it is difficult to parse the metrics, color code, and main results from the paper. One of key findings is that MSR outperforms other techniques, but (a) this is not obvious from the results as its not clear what trends to look for in the metrics (a simple up/down arrow would help) (b) no analysis and discussion of other techniques is provided.


**Summary Of The Paper:**

The paper summarizes various types of classification failures and confidence scoring metrics and identifies their shortcomings with respect to the softmax baseline. Extensive experiments on public image benchmarks (CIFAR-10, CIFAR-100, SVHN, CAMELYON-17-Wilds, iWildCam-2020- Wilds, BREEDS-ENTITY-13) indicate that confidence scoring techniques are sensitive to the dataset choice, integer precision, and calibration.


**Summary Of The Review:**

A useful paper offering insightful, novel analysis, but very difficult to follow.

---

> ### Author Response · Authors · 2022-11-15
> **Reply to Reviewer poSv**
>
> Thank you for your time and  feedback. Below we answer to your questions and concerns.
>
> ------
>
> R: The key weakness is that the paper is difficult to follow.
>
> A: Thank you for this feedback. Please see our answers below regarding the changes we implemented to alleviate your concern. To summarize:
> - We added a  new section (G.1) dedicated to empirically confirming the importance of requirements R1-R3 including 2 new experiments.
> - We added an arrow as suggested to Table 1.
> - We added a visualization for Pitfall 1, thus all three described pitfalls are now visualized (P1 -> Figure 4, P2 -> Figure 1, P3 -> Figure 3) which we hope will help readers to follow the paper.
> - We added a “CSF ranking”  version of the main Table (Table 11) to facilitate getting an overview of the provided information.
>
> ------
>
> R: It is not clear how the pitfalls discussed in Section 2 are evaluated or studied in the remainder of the paper. The paper would benefit from greater organization and extraction of key findings and results that support these conclusions.
>
> A: Thank you for pointing out the need to empirically confirm the importance of our requirements R1-R3 stated in Section 2. We added a new section dedicated to this question in the revised paper. Next to an in-depth analysis linking our results to R1-R3 we provide two new experiments to support the claims. Due to space limitations, this Section is provided in Appendix G.1 and linked to from the start of the main results Section 4.3.
>
> ------
>
> R: A number of relevant method definitions are found in the appendix, rather than the main paper. In addition, definitions of techniques is often found after the fact.
>
> A: Thank you for pointing this out. Given the broad scope of the paper involving multiple task formulations, we have no other choice than to provide them in the appendix and link them from the respective points in the paper. However, we agree that meticulous linking and considering the order of provided information is key to avoiding confusion. We double-checked the paper for a clean flow of information. Could you please point us to the specific locations in the paper where you found inconsistencies?
>
> ------
>
> R: In Table 2, a number of results are reported, however, it is difficult to parse the metrics, color code, and main results from the paper. One of the key findings is that MSR outperforms other techniques, but (a) this is not obvious from the results as its not clear what trends to look for in the metrics (a simple up/down arrow would help)
>
> A: We assume this comment refers to Table 1 (main results) since Table 2 already features arrows. As there is only one metric featured in Table we opted for a bold table caption heading: “FD-Shifts benchmark results measured as AURC ∗10^3 (score range:[0, 1000], lower is better)”. Following your suggestion, we now extend this heading by a down-pointing arrow. The color code is also explained in the subsequent sentence in the caption: “The color heatmap is normalized per column and classifier (separately for CNN and ViT) while whiter colors depict better scores.” From the caption, we further link now to a second version of this Table providing rankings of CSFs as measured by AURC so as to facilitate getting an overview over the provided information.
>
> ------
>
> R: (b) no analysis and discussion of other techniques is provided.
>
> A: The focus of this work is to unite previously separated fields in a unified evaluation protocol. Our arguments focus on why this unification is necessary and our results focus on empirically confirming this importance and demonstrating which insights are enabled by this unification. While in this context it is important to show that methods like ConfidNet, Devries, or DeepGamblers do not generalize to new settings, we hope you agree that providing an in-depth analysis of why methods fail on specific data sets and distribution shifts is out of the scope of this work. The justification of our paper does not depend on these analyses, instead, observing the failure modes and inconsistencies is empirical confirmation that the old protocols are misleading. We feel that this paper already features quite an extensive amount of information and consider it fair to leave further analysis of failure modes to future research. However, we consider it crucial to provide a thorough and transparent report on how methods from the literature were re-implemented to ensure that no performance has been lost along the way and ensure a fair comparison. To this end, we provide an extensive reproducibility study for all employed methods in appendix J.

---

> > ### Comment · Reviewer_poSv · 2022-11-18
> > **thank you**
> >
> > Thank you for your additional clarifications and updates to the paper. I was referring to the definitions in Section A of the Appendix, it might be useful to move them to the main manuscript (although I understand there may be not enough space).
> >
> > It would also help to include a list of key contributions at the end of the introduction in the main paper, as well as reference Appendix G in the appropriate sections in the manuscript.
> >
> > I believe the paper would be a very useful contribution to the conference.

---

### Author Response · Authors · 2022-11-15
**Revision statement**

We thank all three reviewers for their time and valuable feedback. Based on this feedback we uploaded a revised version of the paper. Please find our point-to-point replies below. Finally, we would like to disclaim that we fixed a minor mistake in our open-set experiments, affecting 2 out of the 23 scenarios (i.e. columns) in the results table. This leads to minor relative performance changes in these 2 scenarios but does not affect the arguments, insights, or conclusions described in the paper.

---

### Decision · Program_Chairs · 2023-01-20

**Decision:**

Accept: notable-top-5%

**Justification For Why Not Higher Score:**

N/A

**Justification For Why Not Lower Score:**

The paper received high scores, and perhaps more importantly, is on a topic that is of general interest and will serve to improve the state of best-practices in our community.  It would also be OK to bump it down to a spotlight if there is a shortage of slots for oral presentation, but I think it would make a good oral.

**Metareview: Summary, Strengths And Weaknesses:**

This submission is in some sense a position paper, as well as an enumeration of issues in evaluation of image classification, a set of recommendations, and an empirical study with a code base for classifier evaluation.  The reviewers were unanimous in their praise for the contributions, and the message of taking a step back and understanding common pitfalls in the current state of the literature.  Reviewers were of this opinion that this is a top quality submission, though there were consistent concerns about the (self-contained) presentation.  Some concrete suggestions were discussed between the reviewers and authors, with some potential fixes, though there is acknowledgement that it might be difficult within space limitations.  It was recommended for highlighting within the conference by Reviewer JKia.  A recorded talk/spotlight would provide additional guidance through the paper, which may go some way to making its content more accessible, in addition to bringing the contributions to the attention of a wider audience.

**Note From Pc:**

if the above contains the word "oral" or "spotlight" please see: "oral" presentation means -> notable-top-5% and "spotlight" means -> notable-top-25%. As stated in our emails, we are disassociating presentation type from AC recommendations